ecology/biochemistry/chemical ecology

olfactory communication, volatile organic compounds, gas chromatography-mass spectrometry, solid-phase microextraction, house mouse

**Authors for correspondence:**
Jia Tang
e-mail: jia.tang@ucalgary.ca
Amanda D. Melin
e-mail: amanda.melin@ucalgary.ca

[†]These authors contributed equally.

# Assessing urinary odours across the oestrous cycle in a mouse model using portable and benchtop gas chromatography-mass spectrometry

Jia Tang[1,†], Alice C. Poirier[1,†], Gwen Duytschaever[1], Laís A. A. Moreira[1], Omer Nevo[4,5] and Amanda D. Melin[1,2,3]

[1]Department of Anthropology and Archaeology, [2]Department of Medical Genetics, and [3]Alberta Children's Hospital Research Institute, University of Calgary, Canada
[4]German Centre of Integrative Biodiversity Research (iDiv) Halle-Jena-Leipzig, Germany
[5]Institute of Biodiversity, Friedrich Schiller University Jena, Germany

JT, 0000-0002-6374-0036; ACP, 0000-0001-7947-3721;
GD, 0000-0002-6559-3922; ON, 0000-0003-3549-4509;
ADM, 0000-0002-0612-2514

For female mammals, communicating the timing of ovulation is essential for reproduction. Olfactory communication via volatile organic compounds (VOCs) can play a key role. We investigated urinary VOCs across the oestrous cycle using laboratory mice. We assessed the oestrous stage through daily vaginal cytology and analysed urinary VOCs using headspace gas chromatography-mass spectrometry (GC-MS), testing a portable GC-MS against a benchtop system. We detected 65 VOCs from 40 samples stored in VOC traps and analysed on a benchtop GC-MS, and 15 VOCs from 90 samples extracted by solid-phase microextraction (SPME) and analysed on a portable GC-MS. Only three compounds were found in common between the two techniques. Urine collected from the fertile stages of the oestrous cycle had increased quantities of a few notable VOCs compared with urine from non-fertile stages. These VOCs may be indicators of fertility. However, we did not find significant differences in chemical composition among oestrous stages. It is possible that changes in VOC abundance were too small to be detected by our analytical methods. Overall, the use of VOC traps combined with benchtop GC-MS was the more successful of the two methods, yet portable GC-MS systems may still have utility for some *in situ* applications.

# 1. Introduction

Communicating the timing of ovulation is essential for successful reproduction in many mammalian species. For animals with narrow windows of fertility, such as female mice (*Mus* spp.), which are fertile for only *ca* 24 h, timely communication is especially important [1]. Due to their low-light and solitary habits, many nocturnal rodents rely heavily on chemical signalling to inform socio-sexual behaviours. Volatile organic compounds (VOCs) found in bodily excretions, such as urine, are probably involved in communication between conspecifics [2]. Both male and female mice and rats deposit urine marks in their environment, investigate marks from the other sex, and often deposit over-marks or counter-marks [3]. In females, contact with male rodent urine has been shown to accelerate sexual maturation (i.e. the 'Vandenberg effect') [4–6], induce oestrus (i.e. the 'Whitten effect') [7,8] and terminate the pregnancy (i.e. the 'Bruce effect') [9,10]. Contact with female urine has been shown to increase testosterone levels, accelerate sexual maturation and enhance aggressivity and sexual behaviour in males [11–13]. These studies clearly indicate a role of urinary olfactory cues in mice sexual systems.

A short reproductive cycle length and the importance of olfaction in rodent communication [1] makes female mice a well-suited model for investigating the changes in VOCs across the reproductive cycle [14]. Mice are fertile during the late pro-oestrus and oestrus stages of their oestrous cycle, which is tightly regulated by ovarian hormones [15]. Cycles are recurring and on average 4–5 days long, although 6-day cycles are not uncommon [16]. Variables such as light exposure, age and social grouping can alter cycle regularity and duration [16–18]. For instance, animals housed in laboratories with 12 L : 12 D h cycles have different average cycle lengths compared with animals housed in 14 L : 10 D h cycles [18]. Additionally, females housed in the same cage show puberty delay, cycle suppression and cycle length irregularity when compared with individually housed females [16,17,19]. Given the sensitivity and responsiveness of mice to a combination of abiotic and biotic factors that can alter the window of reproductive opportunity, it may be difficult for males to predict fertile periods. VOCs emitted by female mice during fertile periods may reveal clues about reproductive status to prospective males and facilitate successful reproduction through attracting mates. A number of volatile and semi-volatile chemical indicators of reproductive status in mouse urine have been identified [20–22]. Andreolini *et al.* [20] reported 28 compounds in the urine of female mice using gas chromatography-mass spectrometry (GC-MS). Of these compounds, 11 volatiles varied in concentration across different stages of the oestrous cycle. Nevertheless, the description of biologically active olfactory signals involved is still limited, in part due to the chemical complexity of the VOCs involved in mammalian communication [23].

GC-MS combines the features of both analytical methods, making it possible to identify the structure and relative abundance of unknown VOCs in a sample [24]. Odorous samples, most commonly urine, faeces, body odour or scent gland secretions, are usually collected via swabs rubbed on the gland or body (e.g. anogenital gland secretions of giant pandas, *Ailuropoda melanoleuca* [25]), scent traps such as thermal desorption tubes (e.g. body odour of meerkats, *Suricata suricatta* [26]) or directly into a container (e.g. mouse urine [27,28]), and subsequently transported to a laboratory for analysis on a benchtop GC-MS. More recently, several chromatography companies have begun developing miniaturized GC-MS devices, which can potentially conduct non-invasive and real-time chemical analyses of VOCs in remote conditions. These devices were originally developed for screening specific volatiles in the fields of environmental science [29,30], food manufacturing [31] and chemical warfare [32], but are now starting to be used in the field of animal chemosignalling. For example, the Inficon Hapsite® portable GC-MS device was used for the analysis of the body odour of common marmosets, *Callithrix jacchus*, in both captive [33] and wild [34] conditions. Similarly, Poirier *et al.* [35] used the PerkinElmer Torion® portable GC-MS to analyse swab samples from wild emperor tamarins, *Saguinus imperator* and Weddell's saddleback tamarins, *Leontocebus weddelli*. These portable GC-MS devices, though less sensitive than benchtop systems, may represent a viable alternative to laboratory-based methods for the analysis of samples under field conditions. However, the utility of these portable devices is still unclear and research using them is in its infancy. In particular, portable devices have not yet been employed for the analysis of urinary VOCs; nor have they been used for repeated analysis of odours over time, such as across the female reproductive cycle, or directly compared with benchtop models using the same samples. These comparisons will help assess the utility of different equipment. In the present study, we contribute new data to inform this conversation.

In order to address our research goals, we first needed to detect different phases of the oestrous cycle, and to assign fertile versus non-fertile classifications of each mouse's oestrous cycle. We accomplished this first aim through daily vaginal cell cytology assessment. This allows us to also provide new data

on the variability of oestrous among mice kept in identical conditions. These data may reveal sources of variation underlying the reproductive biology of an important model organism. Our subsequent research aims were: (i) to compare the efficiency of thermal desorption of chromatoprobe VOC traps and analysis on a benchtop GC-MS device, with that of direct headspace extraction by solid-phase microextraction (SPME) and analysis on a portable GC-MS device, in detecting and quantifying urinary VOCs, and (ii) to investigate the VOCs found in the urine of female mice during each stage of the oestrous cycle. The chemical classification of these compounds will yield new fundamental knowledge regarding the substrates of communication and possibly sexual selection and provide a valuable comparison with existing research on murine urinary scents. Additionally, our study design seeks to generate new data for the application of portable GC-MS devices to study the reproductive biology of wild animals in the field. Overall, we aim to improve our understanding of the influence of VOCs in mammalian sensory evolution and provide data and methodologies that can be used in future behavioural and chemistry studies.

# 2. Material and methods

## 2.1. Study species

Ten adult female mice (C57B116-E strain) were donated from another concurrent study at the University of Calgary. The mice originated from the Charles River Laboratories (Wilmington, MA) and were approximately 13 weeks old at the time of sample collection. Each female was individually housed in a polypropylene cage at the Clara Christie Center for Mouse Genomics (University of Calgary, Canada) under standard laboratory-controlled conditions of temperature (22–24°C) and light (12 L : 12 D h cycles). Pelleted feed (Pico-Vac™ Mouse Diet 20) and water were provided ad libitum.

## 2.2. Tracking the oestrous cycle

Visual observation of the external genitalia, electrical impedance measurements of the vaginal epithelial cell layer and evaluation of vaginal cytology are common methods used in oestrous stage identification [36,37]. Vaginal cytology uses the differences in cell type and morphology caused by variation in ovarian hormone levels to distinguish between oestrous stages [15]. This method is the most labour intensive but is required when the different stages of the oestrous cycle need to be accurately identified, which was of interest in our study [1,37]. The division of the oestrous cycle into the four main stages of pro-oestrus, oestrus, metoestrus and dioestrus, is defined by the presence, proportion, density and arrangements of four basic cell types. Collection of the samples can be reliably done either by vaginal lavage or swabbing [1] and both methods have been commonly used [14,15,37]. We found that both methods yielded high-quality sample slides that contained all of the characteristic cells of the vaginal epithelium necessary to assign the oestrous stage (electronic supplementary material, figure S1). We initially started with the swab method, as it was slightly less technically challenging, but we switched to the lavage method as it was less invasive and we found, consistent with Cora *et al*. [1], that lavage yielded a higher cellularity sample than samples collected by swabbing the vagina, thus making assignment of stages easier.

To determine the oestrous stage, we collected one vaginal sample from each mouse for 17 days at approximately 13.00 (electronic supplementary material, table S1). Vaginal smears were collected onto microscope slides using saline swabs (days 1–2) or saline lavage (days 3–17). With the swab method, cotton swabs dipped in saline solution were inserted 1–2 mm into the vaginal canal. With the lavage method, approximately 0.1 ml of saline solution drawn into a pipette was inserted into the vaginal orifice at a depth of 1–2 mm. Care was taken to avoid cervical stimulation during both methods (saline swabs and saline lavage), as this can lead to pseudopregnancy, which appears as prolonged dioestrus for up to 14 days [1,15]. The induction of pseudopregnancy by cervical stimulation is unlikely when samples are collected with care [15]. Our results indicate regular cycling for the entire duration of the study, and we see no evidence of prolonged dioestrus (electronic supplementary material, table S1). Smears were air-dried and stored at room temperature for approximately two weeks before they were stained and mounted using crystal violet and glycerol, respectively. Each slide was viewed using the Zeiss Axio Vert.A1 light microscope at 100× magnification.

We used the presence and ratio of common cell types (i.e. nucleated epithelial cells, cornified epithelial cells and neutrophils) to identify oestrous phases following standard practices [36,37].

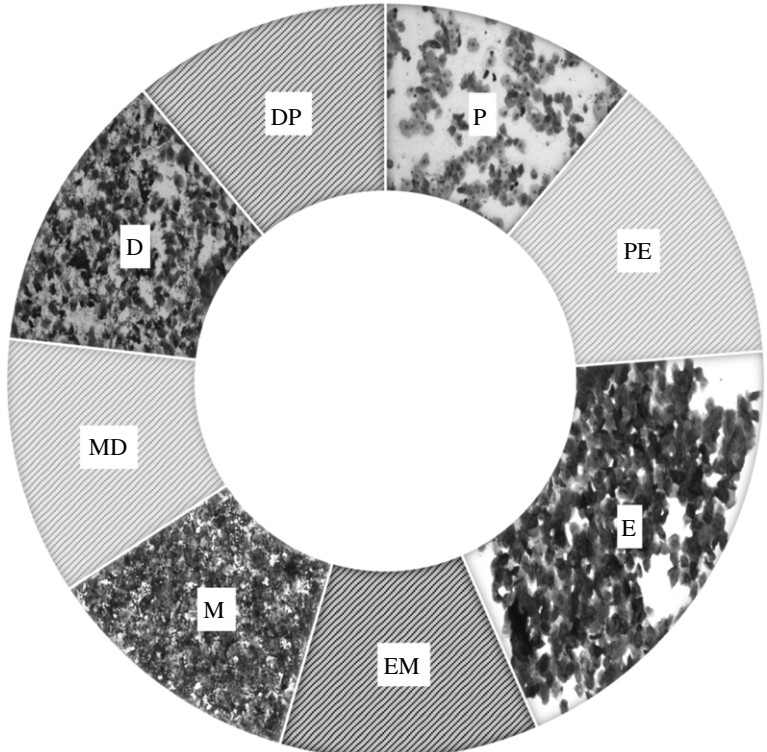

**Figure 1.** Female mice vaginal smears viewed at 100× magnification. Pro-oestrus (P) is characterized by predominantly nucleated epithelial cells. Oestrus (E) is characterized by predominantly cornified epithelial cells. Metoestrus (M) is characterized by the appearance of neutrophils. Dioestrus (D) is characterized by a decreased proportion of cornified epithelial cells and neutrophils. The transitional stages pro-oestrus–oestrus (PE), oestrus–metoestrus (EM), metoestrus–dioestrus (MD) and dioestrus–pro-oestrus (DP) show characteristics of both prior and later stages.

Pro-oestrus smears were predominantly composed of nucleated epithelial cells (figure 1). Smears that consisted almost exclusively of cornified epithelial cells were recorded as oestrus stage (figure 1). Smears dominated by cornified epithelial cells and neutrophils were recorded as metoestrus (figure 1). Lastly, smears that consisted of nucleated epithelial cells, cornified epithelial cells and neutrophils were recorded as dioestrus (figure 1). The oestrous stage of each smear was further categorized and aligned using knowledge of the preceding and succeeding days' stages for accurate identification of the cycling status [1]. This method was especially useful for assessing cycle stages for animals that were sampled during intermediary or transition stages. For instance, females transitioning from pro-oestrus to oestrus will contain both nucleated and cornified epithelial cells and were recorded as pro-oestrus–oestrus stage. To better represent transitional periods, we classified all vaginal smear samples into the following eight oestrous stages: pro-oestrus (P), pro-oestrus–oestrus (PE), oestrus (E), oestrus–metoestrus (EM), metoestrus (M), metoestrus–dioestrus (MD), dioestrus (D) and dioestrus–pro-oestrus (DP). The average length of an oestrous stage was approximated by the oestrous stage recorded at the time of collection. For example, if two oestrus smears were observed on consecutive days, the length of oestrus was counted as 2 days. If a transitional stage smear (e.g. PE) was observed for only 1 day but not on the following day, the length of the transition stage was counted as 1 day.

## 2.3. Analysis of urinary odours

We collected urine in 4 ml glass screw-top vials fitted with a polytetrafluoroethylene/silicone septum (Supelco) following the collection of vaginal smears. Mice were placed onto metal cages and allowed to grasp onto the cage bars. One hand applied gentle pressure to the caudal area of the mouse as the free hand held a vial beneath the mouse to allow the voided urine to be directly collected into the vial [38]. Due to delays caused by the portable GC-MS needing to undergo servicing, we stored the vials prior to analysis at −20°C for an average of 129.2 ± 37.3 days (mean ± s.d.; range 100–205). Importantly, samples collected from different oestrous stages were treated with similar storage conditions which limit potential bias due to different treatments. The volume of urine contained in each vial varied between

mice (9.6 ± 8.0 drops; range 2–40). We decided against standardizing urine volume prior to analysis as it would have led to the unnecessary loss of VOCs upon the opening of each vial and possible exposure to contaminants from instruments and the environment during transfer. Our sampling procedure is anticipated to be robust to differences in sample volume because when samples are allowed to equilibrate and are measured at the same standardized temperature (i.e. 21°C), vapour pressure should not vary with sample volume [39]. To examine this, especially as there was a wide range of urine volume sampled (i.e. a 20-fold variation), we ran regressions of sample volume against the number of VOCs and their relative abundance, and verified the lack of a relationship between sample volume and these variables (electronic supplementary material, figure S2).

Urinary VOCs were then analysed using two different methods: (i) desorption onto a portable GC-MS (PerkinElmer Torion® T-9) following extraction using a SPME fibre and (ii) desorption onto a benchtop GC-MS (Agilent) following adsorption by chromatoprobe VOC traps. Our portable GC-MS needed to be serviced in the middle of our study, when we began measuring samples collected after day 5 of mouse sampling. As a consequence, and due to time constraints, we were not able to analyse all of the urine samples collected. When a portable GC-MS was returned, we prioritized the samples collected more recently, i.e. at the end of the mouse sampling, while trying to the best of our ability to sample across all oestrous stages for each mouse. In total, we analysed 90 samples by portable GC-MS (electronic supplementary material, table S1). We further analysed a representative subset of these samples (i.e. 40 samples) by benchtop GC-MS to compare the methodologies for chemical analyses (electronic supplementary material, table S1).

### 2.3.1. Portable gas chromatography-mass spectrometry

We analysed a total of 90 urine samples using headspace extraction on the PerkinElmer Torion® T-9 portable GC-MS instrument [40] at the University of Calgary, AB, Canada (electronic supplementary material, table S1). Urine samples were defrosted at room temperature for approximately 15 min before they were transferred to a Lab Armor Bead Bath at 37.5°C for 2 min. We then exposed a 65 µm PerkinElmer Custodion® polydimethylsiloxane/divinylbenzene (PDMS/DVB) SPME fibre [40] to the headspace of each sample for 2 min, through the vial septum (vials were not opened, thereby avoiding the loss of VOCs). PDMS/DVB is a generalist combination of absorbents designed to trap a large range of VOCs. The SPME fibre was then desorbed into the injection port of the Torion® GC, fitted with a deactivated 0.048 in i.d. liner, at 270°C for 5 s. The Torion® GC system was fitted with a custom small diameter, low polarity MXT-5 low thermal mass capillary column (5 m × 0.1 mm × 0.4 µm) bundled with electrical resistive heating, allowing for rapid heating and cooling speeds [40]. We used helium at a flow of 0.2 ml min$^{-1}$ as carrier gas (disposable cartridges, PerkinElmer). The split mode was applied at 10 : 1 at 0 s and then 50 : 1 at 10 s, as recommended by the manufacturer [41]. Mass separation was performed by a toroidal ion trap MS on electron ionization mode at 70 eV [40], scanning between 41 and 500 Da. Partway through the experiments, we needed to use a second Torion® instrument when the first underwent repair (electronic supplementary material, table S1). The performance was not detectably different between machines. We tested two different run times, one longer (190.0 s) and one shorter (170.0 s; electronic supplementary material, table S1) to assess the impact on efficacy. In each case, the column temperature began at 50°C (held for 10 s) before it was increased at 2°C s$^{-1}$ until the end temperature of either 290°C (held for 60 s) or 270°C (held for 50 s). Temperature and run times were based on advice from technical experts at PerkinElmer and consistent with product recommendations [41]. We experimented and found that in our trials, lengthening the extraction time beyond the times reported here did not increase the number of compounds detected. Increasing the injection temperature had a negative effect on the SPME fibre, causing burning. We ran Torion® system blanks (i.e. run with nothing injected to ensure the column, injector and ion trap were clean) and SPME fibre blanks (i.e. clean fibre run in the same conditions to control for carry-over) between urine samples. Control samples containing possible contaminant volatiles found in mouse enclosures, such as cage bedding, were also analysed using the same conditions.

### 2.3.2. Benchtop gas chromatography-mass spectrometry

We selected a subset (40) of the urine samples that were analysed on the portable GC-MS, for benchtop analyses using an Agilent GC-MS system (electronic supplementary material, table S1). We selected samples that contained more than five drops of urine for adsorption using chromatoprobe VOC traps, as we could more reliably transfer samples that contained larger volumes of urine via pipette from

the vials into the sealed sampling bag. Urinary VOCs were sampled using a semistatic headspace procedure using chromatoprobe VOC traps, similar to that described by Nevo *et al.* [42,43]. Each chromatoprobe was made of a quartz tube (30 mm long, 3 mm in diameter) and contained 4.5 mg of three adsorbent media (Tenax TA, Carbotrap and Carbosieve S-III, Sigma-Aldrich) in equal amounts (1.5 mg each) trapped between layers of glass wool. A small volume of urine (i.e. two drops sampled with a glass Pasteur pipette, *ca* 0.44 ml) was heated to 37°C and placed into a sealed sampling bag (25 × 38 cm, Toppits), which was attached at one end to a chromatoprobe mounted onto a Teflon tube which was connected to a membrane pump. Samples were left in the sampling bag for 60 min before the air in the bag was suctioned into the VOC trap at 1000 ml min$^{-1}$ for 10 min. Control samples (i.e. empty bags) were additionally collected in the same conditions. The chromatoprobes were shipped on ice to Ulm University, Germany, where they were stored at −20°C for an average of 450 days until analysis. The absorbent materials used in the chromatoprobes, in particular Tenax and Carbosieve S-III, have a high and stable affinity for VOCs, which promotes high shelf stability [44].

The VOCs from the chromatoprobe samples were analysed using thermal desorption and an Agilent GC 7890B equipped with a DB-WAX polar capillary column (30 m × 0.25 mm × 0.25 µm, Agilent) and a cold injection system (CIS4, Gerstel), coupled with an Agilent MS 5977A. Samples were introduced to the thermal desorption unit (TDU) at 30°C in splitless mode, using helium as the carrier gas at a flow rate of 1 ml min$^{-1}$. A full GC-MS methods file is provided as an example analysis protocol in the electronic supplementary material, table S2. After 1 min, the TDU started heating up at 100°C min$^{-1}$ until it reached 250°C, held for 1 min. The liner (Gerstel 6817-U glass liner filled with silanized glass wool) was cooled to −100°C using liquid nitrogen. After the transfer to the liner, it was heated up at 12°C min$^{-1}$ until the temperature reached 250°C, held for 8 min. The initial oven temperature was 40°C, which was maintained for 1 min and then increased by 8°C min$^{-1}$ to 240°C, held for 30 min. The MS transfer line temperature was set to 250°C, the MS source temperature to 230°C and the MS quadrupole temperature to 150°C. The MS operated on electron ionization mode and scanned between 35 and 450 Da.

## 2.4. Classification and identification of volatile organic compounds

Due to differences in analytical methods and characteristics of the two devices, we inspected urine VOC profiles produced using the Agilent benchtop instrument and those using the Torion® portable instrument as separate datasets. Typical examples of chemical profiles obtained using the two instruments are shown in figure 2*a,b*, respectively. For each profile, we performed automatic peak detection, deconvolution and integration using the automated mass spectral deconvolution and identification system (AMDIS 2.73) [45]. The deconvolution parameters used for chemical profiles produced with the benchtop instrument were medium resolution, low sensitivity and high peak shape requirement; those used for profiles produced with the portable instrument were medium resolution, very high sensitivity and medium peak shape requirements. These parameters were optimized for each system, as the benchtop system was more sensitive, and the peaks detected were better resolved, than with the portable system. We removed compounds whose identity was clearly inorganic, e.g. silane derivatives. From the remaining peaks, we only selected those with a minimum area of 0.01% of the chromatogram's total signal. This step allowed us to limit the inclusion of background noise, as peaks under this threshold were generally too flat to be distinguished either from the baseline or from a neighbouring peak. In addition, we removed from further analysis all peaks found in higher amounts in at least one blank sample.

We identified VOCs on the basis of their retention times and mass spectra using the National Institute of Standards and Technology mass spectral library (NIST14) [46]. In the case of samples analysed on the benchtop instrument, additional assistance in the determination of VOC identity was given by their retention indices, which were calculated based on an n-alkane reference mixture analysed under identical conditions. We retained 65 VOCs from samples analysed on the Agilent benchtop instrument and 15 VOCs identified from samples analysed on the Torion® portable instrument. We additionally performed a targeted reverse library search in AMDIS to search for these selected VOCs in all the chemical profiles. For all selected VOCs, we calculated the relative peak area (i.e. peak area divided by the sum of all included peak areas × 100) in each sample, to account for variation in absolute abundance that might be due to the amount of urine collected.

## 2.5. Statistical analyses

We conducted all statistical analyses in R v. 4.0.2 [47] operated in RStudio [48]. We investigated differences in the chemical composition of female mice urine samples both in terms of the number of

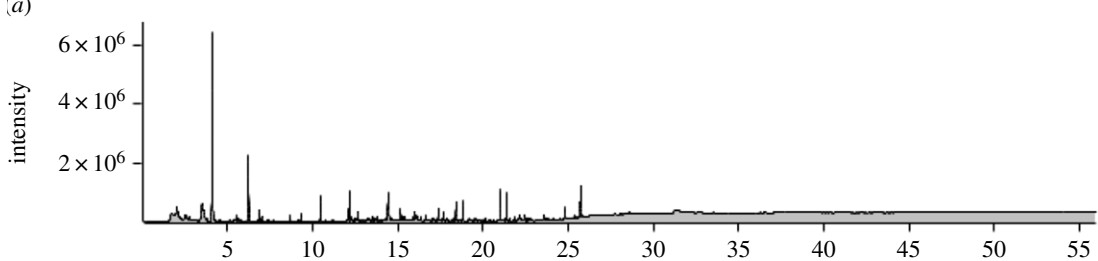

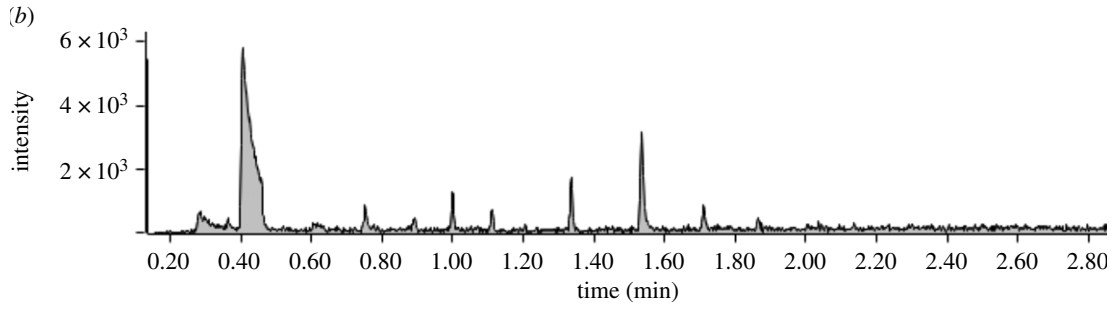

**Figure 2.** Chemical profiles of female mouse urine samples were analysed (*a*) by thermal desorption on the Agilent benchtop GC-MS instrument; (*b*) by SPME on the Torion® portable GC-MS instrument. Note: the scale of the y-axis varies between plots to allow better visualization; the measure of peak intensity differs and is therefore not directly comparable between the two systems.

compounds and their relative abundance in the samples. We tested whether the Torion® portable instrument (i.e. Torion 1 or Torion 2) and the run time (i.e. long or short) used had an influence on the number of compounds retrieved with a generalized linear model (glm function in R base package lme4 v.1.1–23 with Poisson family and log link function). For both benchtop and portable systems, we then assessed the effect of the oestrous stage on the number of compounds detected in the samples using generalized linear mixed models (glmer function in lme4 with Poisson family and log link function). Our full models included the discrete or transitional oestrous stage (i.e. P, PE, E, EM, M, MD, D or DP) as fixed effect, mouse identity (i.e. 1–10, to control for repeated sampling of the same individuals), oestrous cycle ID (i.e. consecutive cycles labelled 1–5, to control for repeated oestrous stages across cycles) and, in the case of the samples analysed on the Torion® portable instrument, run time nested in the instrument, as random effects. We found the variance for all random effects and their interactions to be null, except for oestrous cycle ID in the dataset of samples analysed on the portable instrument. Therefore, we retained as final models (i) the fixed effect oestrous stage (fitted as a generalized linear model, glm function in stats v. 4.0.2) for samples analysed on the benchtop instrument and (ii) the fixed effect oestrous stage and the random effect oestrous cycle ID for samples analysed on the portable instrument. Inspection of model residuals using R package DHARMa v. 0.3.3.0 [49] did not reveal any obvious heteroscedasticity or overdispersion in the data. We applied analysis of variance (ANOVA; Anova function in R package car v. 3.0-1 [50]) and *post hoc* Tukey pairwise tests (glht function in multcomp v. 1.4-14 [51]) to assess the significance of differences between oestrous stages.

We assessed the similarity between chemical profiles of female mice samples collected at different oestrous stages based on pairwise Bray–Curtis dissimilarity indices, calculated from the $\log(x + 1)$-transformed relative peak areas using the vegdist function in R package vegan v. 2.5-6 [52]. For samples analysed on the Torion® portable GC-MS, we ran analyses of similarities (anosim function in vegan with 999 permutations and the Bray–Curtis dissimilarity index) to detect statistical differences in chemical composition among samples run on the two instruments and using the long and short run times. The two portable instruments yielded comparable results in terms of their chemical composition (ANOSIM: $R = -0.08$, $p = 0.91$). Hence, we combined samples analysed using Torion 1 and Torion 2. However, run time significantly affected sample chemical composition ($R = 0.19$, $p < 0.01$). We consequently used run time as a grouping factor in our multivariate chemical analyses.

We performed a multifactorial permutational multivariate analysis of variance using distance matrices (PerMANOVA; adonis2 function in vegan) to assess the relative effect of oestrous stage, mouse identity (to control for repeated sampling of the same individuals) and oestrous cycle ID (to

control for repeated oestrous stages across cycles) on sample chemical composition. The adonis2 function allowed testing the effect of one predictor while accounting for the effects of other predictors, with the arg = 'margins' argument. We used 999 permutations, the Bray–Curtis dissimilarity index and, in the case of the samples analysed on the portable instrument, run time as a strata argument within which to constrain permutations. Unlike most statistical models, the only statistical assumption of PerMANOVA is to ensure multivariate homogeneity of variance within each group tested. This assumption was verified using the permutation test for homogeneity of multivariate dispersion (permutest function in vegan, using 999 permutations), on the measure of group multivariate homogeneity of variance computed using the betadisper function in vegan.

Finally, we inspected the variation in relative peak area across the oestrous cycle for each identified compound in both datasets (i.e. 65 compounds for samples analysed on the benchtop instrument and 15 compounds for samples analysed on the portable instrument), using Kruskal–Wallis rank-sum tests (kruskal.test function in stats) and associated *post hoc* Dunn's tests (dunnTest function in FSA v. 0.8.30 [53]). Because many compounds were only detected in a subset of oestrous stages, the power of the tests was low. Therefore, we further grouped the eight oestrous stages into two fertility states, with stages P, PE and E as the 'fertile' state, and stages EM, M, MD, D and DP as the 'non-fertile' state. We tested differences in VOC relative peak area between fertile and non-fertile states for each compound using Wilcoxon rank-sum tests (wilcox.test function in stats).

# 3. Results

## 3.1. Study of oestrous cyclicity

Examining vaginal smears collected from 10 mice over 17 consecutive days revealed that the mice were not cycling in unison with each other. The length of a full oestrous cycle varied considerably between mice and ranged 3–6 days (electronic supplementary material, table S1). The average length of a full oestrous cycle was $4.7 \pm 0.8$ (mean ± s.d.) days. Pro-oestrus lasted on average for $1.0 \pm 0.2$ days. Oestrus was the longest and most variable in length ($1.8 \pm 0.7$ days). Metoestrus was the shortest and least variable in length ($1.0 \pm 0.1$ days). Dioestrus lasted $1.1 \pm 0.3$ days. Oestrus was the most frequently captured stage (58 days across all mice), followed by dioestrus (30 days), metoestrus (28 days) and pro-oestrus (21 days; electronic supplementary material, table S1).

## 3.2. Detection and identification of mouse urinary volatile organic compounds

We detected 65 VOCs in 40 chromatoprobe samples analysed on the Agilent benchtop GC-MS (table 1) and 15 VOCs in 90 vial samples extracted by SPME and analysed on the Torion® portable GC-MS (table 2). Samples analysed on the benchtop device contained a mean ± s.d. of $38.7 \pm 6.5$ VOCs (range 22–49), those analysed on the portable device contained $3.4 \pm 1.7$ VOCs (range 0–7). One of the portable GC-MS devices yielded a higher number of VOCs (Torion 1: $3.5 \pm 1.6$; Torion 2: $2.8 \pm 1.9$), although this difference was not significant (GLM: $Z = -1.42$, $p = 0.16$). The shorter run time led to the detection of a slightly higher number of VOCs ($3.4 \pm 1.7$) than the longer run time ($3.3 \pm 1.7$), though this was not significantly different ($Z = -0.33$, $p = 0.74$).

We assigned tentative identities to 62 of the 65 VOCs detected in samples analysed by thermal desorption on the Agilent benchtop instrument (table 1), and 11 of the 15 VOCs detected in samples analysed by SPME on the Torion® portable instrument (table 2). The identified VOCs were mainly hydrocarbons, alcohols, aldehydes and ketones, sometimes containing an aromatic group, most of which have been reported in the mammalian semiochemistry literature. Only three VOCs were reliably detected in the samples by both systems: butan-2-one, pentan-2-one and 3,4-dehydro-exo-brevicomin.

## 3.3. Variation in the chemical composition of female mouse urine across the oestrous cycle

The 40 urine probe samples analysed on the Agilent benchtop instrument spanned seven oestrous stages (no sample collected during the EM stage was analysed) in the 10 mice; the 90 urine samples analysed on the Torion® portable GC-MS spanned all eight oestrous stages. We observed some variation in the number of compounds retrieved from the samples across the oestrous cycle (figure 3), although the differences between oestrous stages were not significant (ANOVA benchtop GC-MS: $\chi^2 = 8.42$, d.f. = 6,

**Table 1.** List of 65 VOCs identified from female mouse urine samples analysed by thermal desorption on an Agilent benchtop GC-MS instrument. VOCs in italics were also found in the dataset analysed on the portable instrument; VOCs highlighted in bold were present in higher abundance in samples collected during the fertile phase of the oestrous cycle. We also provide references to the literature where the same compound has been detected in other studies on mammalian urinary semiochemistry.

| VOC no. | retention time (mean ± s.d.) | tentative name | mol. weight (g mol⁻¹) | presence in oestrous stages (EM not sampled) | number of samples | references on mammalian urinary semiochemicals[a] |
|---|---|---|---|---|---|---|
| 01 | 1.79 ± 0.01 | pentane | 72.2 | P, E, MD | 7 | [54] |
| **02** | **1.92 ± 0.01** | **pent-1-ene or cyclopentane** | **70.1** | **P, PE, E, M, D, DP** | **27** | pent-1-ene: **[54]** |
| 03 | 2.54 ± 0.01 | propan-2-one | 58.1 | all stages | 40 | [54] |
| 04 | 3.04 ± 0.01 | ethyl acetate | 88.1 | P, E, M, DP | 5 | [55] |
|  |  |  |  |  |  | [56] |
| 05 | 3.15 ± 0.01 | *butan-2-one* | 72.1 | P, PE, E, M, D, DP | 16 | [57–62] |
|  |  |  |  |  |  | [54,55,63] |
|  |  |  |  |  |  | [56,64–66] |
| 06 | 3.78 ± 0.01 | branched C9 alkane | unk. | P, PE, E, M, D, DP | 13 | n/a |
| 07 | 3.98 ± 0.01 | *pentan-2-one* | 86.1 | P, PE, E, M, D, DP | 19 | [28,57–62] |
|  |  |  |  |  |  | [54,55,63,67] |
|  |  |  |  |  |  | [56,64–66,68] |
|  |  |  |  |  |  | [69] |
|  |  |  |  |  |  | [70] |
| 08 | 4.26 ± 0.00 | decane | 142.3 | all stages | 35 | [67,71,72] |
|  |  |  |  |  |  | [69] |
| 09 | 4.57 ± 0.01 | α-pinene | 136.2 | all stages | 38 | — |

(Continued.)

| VOC no. | retention time (mean ± s.d.) | tentative name | mol. weight (g mol⁻¹) | presence in oestrous stages (EM not sampled) | number of samples | references on mammalian urinary semiochemicals[a] |
|---|---|---|---|---|---|---|
| 10 | 4.79 ± 0.01 | methylbenzene | 92.1 | P, E, M, D, DP | 13 | [61,73] / [54] |
| 11 | 5.25 ± 0.01 | unidentified 1 | unk. | PE, E, M, D | 9 | n/a |
| 12 | 5.52 ± 0.01 | undecane | 156.3 | all stages | 35 | [27,28] / [71] |
| 13 | 6.82 ± 0.07 | branched C12 alkane | unk. | all stages | 38 | n/a |
| 14 | 6.98 ± 0.06 | D- or L-Limonene | 136.2 | all stages | 36 | [74,75] / [55] / [64,65,76] / [69] |
| 15 | 7.34 ± 0.04 | eucalyptol | 154.2 | P, PE, E, M, D, DP | 16 | — |
| 16 | 8.61 ± 0.11 | branched C13 alkane | unk. | all stages | 38 | n/a |
| 17 | 8.54 ± 0.02 | **1,2,3- or 1,2,4-Trimethylbenzene** | **120.2** | **P, PE, E, M, D** | **10** | **[65,68]** |
| 18 | 8.69 ± 0.04 | octanal | 128.2 | PE, M, D | 5 | [57,59–61] / [54,63,67,71,72,77] / [56,65,76] |
| 19 | 10.36 ± 0.09 | branched C14 alkane | unk. | P, PE, E, M, D, DP | 28 | n/a |
| 20 | 10.46 ± 0.04 | nonanal | 142.2 | all stages | 40 | [26,58–61] / [54,55,63,67,71,72,77–80] / [64,65] / [81] |

(Continued.)

**Table 1.** (Continued.)

| VOC no. | retention time (mean ± s.d.) | tentative name | mol. weight (g mol⁻¹) | presence in oestrous stages (EM not sampled) | number of samples | references on mammalian urinary semiochemicals[a] |
|---|---|---|---|---|---|---|
| 21 | 10.79 ± 0.01 | *3,4-dehydro-exo-brevicomin* | 154.2 | all stages | 18 | [28,73,82] |
| 22 | 11.18 ± 0.01 | 1,2,4,5-tetra methylbenzene | 134.2 | P, PE, E, M, MD, D | 10 | [65] |
| 23 | 12.09 ± 0.01 | 2-ethylhexan-1-ol | 130.2 | all stages | 40 | [60,61] |
| 24 | 12.18 ± 0.02 | decanal | 156.2 | all stages | 37 | [27,59–62]  [55,67,71,72,77]  [64,76]  [69]  [81] |
| 25 | 12.62 ± 0.01 | benzaldehyde | 106.1 | P, PE, E, M, D, DP | 37 | [20,57,59–62,73,83]  [54,55,63,67,71,72,77,78,80,82]  [64,68]  [81] |
| 26 | 13.32 ± 0.05 | caryophyllene or longifolene | 204.4 | all stages | 36 | caryophyllene: [28,73] |
| 27 | 13.67 ± 0.02 | hexadecane | 226.4 | P, PE, E, M, D, DP | 15 | [61] |
| 28 | 13.81 ± 0.02 | undecanal | 170.3 | all stages | 38 | [62]  [72]  [81] |

(*Continued.*)

**Table 1.** (Continued.)

| VOC no. | retention time (mean ± s.d.) | tentative name | mol. weight (g mol⁻¹) | presence in oestrous stages (EM not sampled) | number of samples | references on mammalian urinary semiochemicals[a] |
|---|---|---|---|---|---|---|
| 29 | 14.57 ± 0.01 | acetophenone | 120.2 | P, PE, E, M, D | 25 | [19,26,27,57,60–62,73,74,83] [63,67,71,72,77,82,84,85] [64,65,68,76] [70] |
| 30 | 14.66 ± 0.01 | nonan-1-ol | 144.3 | all stages | 29 | [71] [68] |
| 31 | 15.14 ± 0.02 | unidentified 2 | unk. | all stages | 32 | n/a |
| **32** | **15.17 ± 0.01** | **heptadecane** | **198.4** | **P, PE, E** | **10** | **[61,73]** **[76]** |
| 33 | 15.26 ± 0.02 | unidentified saturated cycloalkane | unk. | all stages | 18 | n/a |
| 34 | 15.36 ± 0.02 | dodecanal | 184.3 | all stages | 39 | [27] [63,72] [68,76] [86] |
| 35 | 15.52 ± 0.08 | 3-methylpentanoic acid | 116.2 | PE, E, M, D | 6 | — |
| **36** | **15.92 ± 0.02** | **dioctyl ether** | **242.4** | **all stages** | **29** | — |
| 37 | 16.12 ± 0.01 | decan-1-ol | 158.3 | all stages | 37 | [27] |
| 38 | 16.36 ± 0.01 | N,N-dibutylformamide | 157.3 | all stages | 34 | — |
| 39 | 16.58 ± 0.01 | octadecane | 254.5 | P, E, M, MD | 7 | — |
| 40 | 17.01 ± 0.01 | anethole | 148.2 | all stages | 26 | — |
| 41 | 17.08 ± 0.01 | isopropyl dodecanoate | 242.4 | all stages | 22 | — |

(Continued.)

**Table 1.** (Continued.)

| VOC no. | retention time (mean ± s.d.) | tentative name | mol. weight (g mol⁻¹) | presence in oestrous stages (EM not sampled) | number of samples | references on mammalian urinary semiochemicals[a] |
|---|---|---|---|---|---|---|
| 42 | 17.37 ± 0.01 | geranyl acetone | 194.3 | all stages | 38 | [26,27,57,62,75], [71,77,82], [76] |
| 43 | 17.69 ± 0.01 | unidentified 3 | unk. | all stages | 40 | n/a |
| 44 | 17.98 ± 0.01 | dimethyl sulfone | 94.1 | P, PE, E, M, DP | 13 | [60,61], [67,71,72,87], [56,65,76], [88] |
| 45 | 18.26 ± 0.01 | tetradecanal | 212.4 | P, PE, E, M, MD, D | 22 | [76] |
| **46** | **18.83 ± 0.01** | **dodecan-1-ol** | **186.3** | **All stages** | **40** | **[28], [76]** |
| 47 | 19.17 ± 0.01 | dodecyl prop-2-enoate | 240.4 | PE, E, M | 3 | — |
| 48 | 19.46 ± 0.02 | diphenyl ether | 170.2 | all stages | 28 | — |
| 49 | 19.60 ± 0.01 | pentadecanal | 226.4 | all stages | 34 | [72] |
| 50 | 20.84 ± 0.01 | cedrol or epicedrol | 222.4 | PE, E, M | 4 | — |
| 51 | 20.89 ± 0.01 | hexadecanal | 240.4 | P, PE, E, M, D, DP | 19 | [61], [76] |
| 52 | 21.02 ± 0.01 | 2-phenoxyethanol | 138.2 | all stages | 39 | [72] |
| 53 | 21.33 ± 0.01 | tetradecan-1-ol | 214.4 | all stages | 36 | [89], [71], [76] |

(Continued.)

**Table 1.** (Continued.)

| VOC no. | retention time (mean ± s.d.) | tentative name | mol. weight (g mol⁻¹) | presence in oestrous stages (EM not sampled) | number of samples | references on mammalian urinary semiochemicals[a] |
|---|---|---|---|---|---|---|
| 54 | 21.85 ± 0.01 | n-hexyl salicylate | 222.3 | all stages | 32 | — |
| **55** | **22.84 ± 0.01** | **2-ethylhexyl salicylate** | **250.3** | **P, PE, E, M, D** | **18** | — |
| 56 | 23.62 ± 0.01 | hexadecan-1-ol | 242.4 | all stages | 30 | [71,72] [76] |
| 57 | 24.83 ± 0.01 | benzophenone | 182.2 | all stages | 39 | — |
| 58 | 25.75 ± 0.01 | octadecan-1-ol | 270.5 | all stages | 39 | [89] [71] |
| 59 | 25.85 ± 0.01 | hexacosane | 366.7 | P, PE, E, M, D | 10 | — |
| 60 | 26.88 ± 0.01 | heptacosane | 380.7 | P, PE, E, M, D, DP | 16 | — |
| 61 | 28.08 ± 0.02 | octacosane | 394.8 | P, PE, E, M, D, DP | 14 | [61] |
| 62 | 29.55 ± 0.03 | nonacosane | 408.8 | P, PE, E, M, D | 10 | — |
| 63 | 31.31 ± 0.03 | triacontane | 422.8 | P, PE, E, M | 6 | — |
| 64 | 33.54 ± 0.05 | hentriacontane | 436.8 | PE, E, M | 3 | — |
| 65 | 36.31 ± 0.05 | dotriacontane | 450.9 | PE, E, M | 3 | — |

[a] Rodentia & Lagomorpha; Carnivora; Primates; Artiodactyla; Proboscidea; Marsupials.

**15**

**Table 2.** List of 15 VOCs identified from female mouse urine samples analysed by solid-phase microextraction on the Torion® portable GC-MS instrument. VOCs in italics were also found in the dataset analysed on the benchtop instrument; VOCs highlighted in bold were present in higher abundance in samples collected during the fertile phase of the oestrous cycle. We also provide references to literature where the same compound has been detected in other studies on mammalian urinary semiochemistry.

| VOC no. | retention time (mean ± s.d.) | tentative name | mol. weight (g mol$^{-1}$) | presence in oestrous stages | number of samples | references on mammalian urinary semiochemicals[a] |
|---|---|---|---|---|---|---|
| 01 | 0.29 ± 0.01 | *butan-2-one* | 72.1 | all stages | 50 | [57–62], [54,55,63,67,78], [56,64–66] |
| 02 | 0.36 ± 0.01 | butane-2,3-dione | 86.1 | P, PE, E, EM, M, MD, D | 25 | [28,57–62], [54,55,63,67], [64,65,68,90], [69], [70] |
| 03 | 0.41 ± 0.00 | heptan-4-one or 1,2-dimethylpropyl acetate | 130.2 / 114.2 | all stages | 87 | heptan-4-one: [20,28,60–62,82], [54,55,67,77,78,82,84], [56,64–66,76], [69] |
| 04 | 0.47 ± 0.07 | *pentan-2-one* | 86.1 | P, PE, E, M, D, DP | 16 | [28,57–61], [54,55,63,67,77], [56,64–66,68], [69], [70] |
| 05 | 0.48 ± 0.01 | unidentified (containing N) | unk. | P, E, M, DP | 7 | n/a |

(Continued.)

**Table 2.** (Continued.)

| VOC no. | retention time (mean ± s.d.) | tentative name | mol. weight (g mol⁻¹) | presence in oestrous stages | number of samples | references on mammalian urinary semiochemicals[a] |
|---|---|---|---|---|---|---|
| 06 | 0.60 ± 0.01 | unidentified (containing N) | unk. | P, E, EM, M, MD, D | 14 | n/a |
| 07 | 0.64 ± 0.00 | unidentified (containing S) | unk. | E, M | 3 | n/a |
| **08** | **0.66 ± 0.00** | **butanoic acid** | **88.1** | **E** | **3** | **[28,58,59]** **[71,72,84]** **[90]** |
| 09 | 0.68 ± 0.01 | 2-methylbutanoic acid or 2-methyl pentanoic acid | 102.1 | E, M, D | 4 | 2-methyl butanoic acid: [28,58,59] [72] [76,90] 2-methyl pentanoic acid: [55] |
| 10 | 0.71 | unidentified $C_7H_{14}N_2$ compound | unk. | D | 1 | n/a |
| 11 | 0.75 ± 0.00 | 4- or 5-hepten-2-one | 112.2 | all stages | 40 | 4-hepten-2-one: [20,73] 5-hepten-2-one: [19,60,61,73,74] |
| **12** | **1.01 ± 0.07** | **3,4-dehydro-exo-brevicomin** | **154.2** | **all stages** | **44** | **[28,73,82]** |
| 13 | 1.01 | unidentified | unk. | M | 1 | n/a |
| 14 | 1.22 | benzene acetaldehyde (or other $C_{12}H_{16}O$) | 120.2 | M | 1 | [68,76] |
| 15 | 1.37 ± 0.00 | unidentified C9 alkene | unk. | E, M, MD, D | 8 | n/a |

[a] Rodentia & Lagomorpha; Carnivora; Primates; Artiodactyla; Proboscidea; Marsupials.

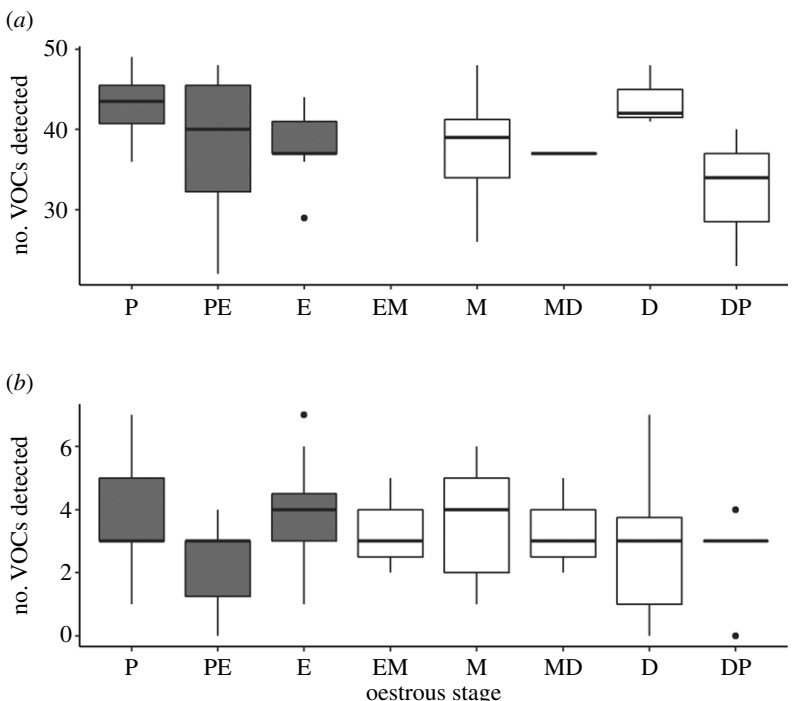

**Figure 3.** Variation across the oestrous cycle of the number of VOCs detected in mouse urine samples analysed (*a*) by thermal desorption on the Agilent benchtop GC-MS (*n* = 40, no sample collected at the EM stage), and (*b*) by SPME on the Torion® portable GC-MS (*n* = 90).

**Table 3.** Results of PerMANOVA comparing the effect of oestrous stage, mouse identity and oestrous cycle ID on the chemical composition of urine samples analysed on the Agilent benchtop instrument and on the Torion® portable instrument. *p*-values are significant at $p \leq 0.05$ (in italics).

| GC-MS technique (*n* samples) | factor | d.f. | sum of squares | $R^2$ | pseudo-*F* | *p* |
|---|---|---|---|---|---|---|
| benchtop GC-MS (*n* = 40) | oestrous | 6 | 0.33 | 0.14 | 0.90 | 0.59 |
| | mouse | 9 | 0.53 | 0.22 | 0.97 | 0.53 |
| | cycle ID | 3 | 0.23 | 0.10 | 1.24 | 0.22 |
| | residual | 21 | 1.28 | 0.54 | | |
| portable GC-MS (*n* = 90) | oestrous | 7 | 0.38 | 0.06 | 0.79 | 0.73 |
| | mouse | 9 | 0.89 | 0.14 | 1.46 | 0.07 |
| | cycle ID | 4 | 0.78 | 0.12 | 2.88 | *<0.01* |
| | residual | 66 | 4.48 | 0.69 | | |

$p = 0.21$; portable GC-MS: $\chi^2 = 6.57$, d.f. = 7, $p = 0.47$; non-significant *post hoc* Tukey pairwise tests, electronic supplementary material, table S3).

Similarly, our metric of the chemical composition of samples, the Bray–Curtis dissimilarity index, did not significantly differ between oestrous stages (table 3). In the case of samples analysed on the portable GC-MS, the PerMANOVA analysis revealed a greater variation between individual female mice sampled, and between oestrous cycle ID, than between stages of the oestrous cycle (table 3).

We did not find any significant variation in VOC relative peak area across the oestrous cycle (electronic supplementary material, table S4); nevertheless, a few VOCs showed a significant increase of their relative area in the fertile state compared with the non-fertile state, in both samples analysed on the benchtop GC-MS (electronic supplementary material, figure S3), and those analysed on the portable GC-MS (electronic supplementary material, figure S4). On the benchtop instrument, this included pent-1-ene/cyclopentane (#02; Wilcoxon rank exact test: $W = 42$, $p = 0.02$, $n = 27$), 1,2,3-/1,2,4-trimethylbenzene (#17; $W = 23$, $p = 0.02$, $n = 10$), dioctyl ether (#36; $W = 149.5$, $p = 0.01$, $n = 29$), dodecan-1-ol (#46; $W = 271$, $p = 0.02$, $n = 40$)

and 2-ethylhexyl salicylate (#55; $W = 53$, $p = 0.01$, $n = 18$). In addition, heptadecane (#32, $n = 10$) was only detected in samples from the fertile state (electronic supplementary material, figure S3). In samples analysed on the portable instrument, only 3,4-dehydro-exo-brevicomin (#12) showed an increase of its relative area during the fertile state ($W = 336$, $p = 0.03$, $n = 44$). In addition, butanoic acid (#08, $n = 3$) was specific to fertile samples, while the unidentified $C_7H_{14}N_2$ compound (#10, $n = 1$), the unidentified compound #13 ($n = 1$) and benzeneacetaldehyde (#14, $n = 1$) were only found in non-fertile samples (electronic supplementary material, figure S4). However, these compounds were very rare (i.e. found in only 1–3 out of 90 samples).

# 4. Discussion

## 4.1. Variation in mouse oestrous cycles and implications for reproductive biology

We found that the average length of a complete mouse oestrous cycle was $4.7 \pm 0.8$ days. This is consistent with the average oestrous cycle length reported in mice and rats, which averages 4–5 days in length [1,16]. The occasional occurrence of prolonged oestrus stage and absence of other stages resulted in variable (3–6 day) cycle lengths. Cycle length variability and the absence of some stages may be explained by natural variation between individual mice and our sampling frequency, respectively. Oestrous stages can range between 6 and 72 h depending on the stage and mouse [1]. We collected vaginal smears once every 24 h, which could have resulted in missing stages that were shorter than 24 h. However, the rare occurrence of longer than average (6–7 day) cycles is not uncommon, especially for mice, which exhibit greater cycle length variability when compared with rats [16,91]. Inter-animal variability was expected, but it is possible that environmental factors, namely cage proximity to animals belonging to concurrent studies, played a role in oestrous cycle variability.

Overall, vaginal cytology was a successful method for distinguishing between the different stages of each animal's oestrous cycle. However, occasionally, reference to the previous and subsequent stages, in addition to the appearance of vaginal smears, was necessary in confident oestrous stage identification. For example, some oestrous stages were not easily distinguishable due to transitional stages (PE, EM, MD, DP), missing or prolonged stages, and poorly prepared slides. Additionally, it was often difficult to differentiate between metoestrus and dioestrus, which only differed by the relative proportion of the same cell types. Cytological results may be improved by the use of polychromatic stains such as Shorr stain [92] or Papanicolaou stain [91,93], which both provide improved ability to discern the degree of epithelial cell cornification.

## 4.2. Evaluation of different headspace gas chromatography-mass spectrometry techniques for the study of mammalian urinary volatiles and recommendations for future use

The use of the VOC traps combined with conventional benchtop GC-MS was more successful than SPME combined with portable GC-MS. We detected 65 compounds from 40 chromatoprobe VOC traps analysed using the thermal desorption extraction technique on the Agilent benchtop GC-MS instrument, and 15 compounds from 90 samples analysed using the SPME headspace extraction technique on the Torion® portable GC-MS instrument. This was despite the fact that we used more inclusive peak detection parameters on the portable GC-MS VOC profiles. Moreover, only three of the identified compounds, butan-2-one, pentan-2-one and 3,4-dehydro-exo-brevicomin, were found in common using the two techniques. This important difference in sensitivity and affinity for certain VOCs highlights the difficulty in performing comparative analyses of results obtained using different analytical techniques. Poirier [68], who compared the use of SPME on a PerkinElmer Clarus 500 benchtop GC-MS and on a Torion® portable GC-MS for the analysis of wild tamarin scent gland swabs, similarly detected a lower number of VOCs when using the portable instrument. Kücklich et al. [33], who compared the performance of the Hapsite® portable GC-MS with that of cotton swabs analysed on an Agilent 5973 benchtop GC-MS for the analysis of body odour of captive common marmosets, also found that the portable instrument retrieved fewer compounds. The difference observed between benchtop and portable systems in the aforementioned studies, as well as in our study, was expected to some degree, since the extremely short GC run in a portable instrument (i.e. around 3 min in our study, in comparison with the 56 min run time with the benchtop instrument) does not allow complete separation of individual chemicals over time. Moreover, it is possible that the VOC trap was more effective at capturing VOCs from the headspace inside the urine vials than the

SPME fibre. As the fibre coating (i.e. PDMS/DVB) in SPME forms an equilibrium with the substrate and headspace during extraction, only a portion of the VOCs are captured onto the fibre; the affinity of the coating for the different chemicals sampled is variable. With the VOC trap, however, almost all of the VOCs present in the headspace are pumped into the chromatoprobe, regardless of their size and structure. The differences in VOC sampling, extraction and detection capacities between the two systems used here made drawing direct comparisons difficult [94].

To the best of our knowledge, this was the first use of the chromatoprobe VOC traps for the extraction of urinary VOCs. These sampling devices have been successfully used for the collection of VOCs emitted by fruits [42,43], flowers [95] and the human skin [96]. Our results indicate that chromatoprobe VOC traps were successful in the extraction and storage of a large number of diverse urinary VOCs. These chromatoprobes can be self-made and are cheap to produce. They are small and light enough to be easily transported to, and stored at a remote field site, and only require a pump to be used [42,43,97]. Their thermal desorption, however, can only be done on a benchtop GC-MS instrument equipped with a TDU. Chromatoprobe VOC traps can be additionally optimized for the purpose sought by the researchers. The absorbent mixture our probes are composed of (i.e. 1.5 mg Tenax TA, 1.5 mg Carbotrap and 1.5 mg Carbosieve S-III) functions as a general, multi-purpose absorbent tailored to exceptional long-term storage, even under field conditions without access to frozen storage. The absorbent materials have a high and stable affinity for most VOCs, as demonstrated by Woolfenden [98] and Magnusson *et al.* [44] for Tenax and Carbosieve S-III, respectively. As a result, our chromatoprobes should have high shelf stability. In other applications, they did not observably lose signal strength even after more than 3 years and with transport from remote tropical stations and no cooling (O Nevo 2021, unpublished data). Previous research has shown that the volatile components in human urine degrade over time due to evaporation and bacterial activity when stored at 4°C, and more slowly when frozen, even at −80°C [99,100]. Accordingly, loss of VOCs may have occurred from the samples during storage at −20°C, prior to transfer to the chromatoprobes, although the storage at freezing temperatures probably limited this loss. Our study was outside of the tested range of storage conditions for urinary samples [99,100], so we cannot completely predict the impact on loss of highly volatile compounds involved in chemical signalling. Nevertheless, we do not expect systematic biases with respect to our conclusions regarding differences across oestrous stages because samples from different cycle phases were exposed to the same storage conditions. We believe that given the stability of chromatoprobes, the continuous freezing applied to our samples, and similarity of treatment of samples across different oestrous stages, our conclusions are not strongly biased by the storage conditions, but systematic future examination would be useful for revealing the precise impacts of storage conditions on analytical outcomes.

Our study also presented the first use of the Torion® portable GC-MS for the analysis of urinary VOCs. The efficiency of this device appeared limited, yet several methodological aspects may be improved to ensure better results in the future. These include optimizing VOC extraction and detection procedure through the choice of SPME fibre coating, injection parameters, GC column temperature cycle and MS detector parameters. Despite limitations, this instrument or similar devices may still have utility for some *in situ* chemical analyses. In particular, portable GC-MS devices could be efficiently used for an *in situ* search for VOCs which have been identified in prior studies using a benchtop instrument, and for which the detectability using this instrument has already been verified in pilot studies [33]. *In situ* collection and chemical analysis of urinary VOCs could prove useful, since urine is collected non-invasively and relatively easily in wild conditions. It could notably be used for the rapid screening of diseases in free-living animals and in humans [101–103].

## 4.3. Assessment of potential signalling compounds in mouse reproductive communication

We found no significant variation in the chemical composition of female mouse urine samples along the oestrous cycle, neither in the number of VOCs detected, nor in the combination and relative abundance of VOCs as measured by the Bray–Curtis dissimilarity index. This result did not differ between samples analysed by thermal desorption on the Agilent benchtop GC-MS and those analysed by SPME on the Torion® portable GC-MS. However, differences between oestrous stages may have been masked by the many non-oestrous related VOCs also present in mouse urine. Future research on the chemical cues of oestrus in mice could make use of males, pre-reproductive animals, and even neutered females, in order to find and discount all compounds irrelevant to oestrous. It is also possible that the variation in urine VOCs concentration at play during the oestrous cycle was too subtle to be detected by our analytical methods. Our method, headspace GC-MS, was limited to the analysis of exclusively

volatile chemicals in mice urine; we identified VOCs with a molecular weight between 58.1 (propan-2-one) and 450.9 g mol$^{-1}$ (dotriacontane) using VOC trap thermal desorption on the benchtop device; this range was lower when using SPME on the portable device. However, it has been shown that much bigger chemicals, of molecular weight of several kDa such as the pheromone-binding major urinary proteins (MUPs), constitute key scent signalling media in rodents [22,104–106]. The interplay between these MUPs and the VOCs they bind with is likely to represent a more accurate description of the chemical signalling involved in murine sexual communication. Moreover, a study in more naturalistic conditions might have yielded different results, as it is likely that some active compounds are products of chemical degradation that require time or conditions we may not have reached in our experimental settings.

In our study, we observed an effect of oestrous cycle ID on the chemical dissimilarity of urine samples for samples analysed on the portable instrument. This was a surprising result given that all ten female mice were kept in the same conditions across the consecutive oestrous cycles. Yet, because urinary odour samples were not collected every single day of the experiment and oestrous stages varied in their duration, not all oestrous stages were represented for each mouse during the oestrous cycle. This could explain why sample chemical composition did not significantly vary across oestrous stages. A bigger sample size, including more individual female mice sampled at every stage of their oestrous cycle, would be necessary to further investigate potential changes in urine VOCs along with the oestrous cycle.

We identified a total of 77 VOCs in the urine of female mice (three VOCs were common to the two systems used), which was in the range of previous findings: Schwende *et al.* [73] identified 61 VOCs, and Röck *et al.* [59] 70 VOCs, from the urine of house mice. In related species, Soini *et al.* [27] reported 60 VOCs from the urine of mound-building mice, *Mus spicilegus*; Boyer *et al.* [75] 23 VOCs from pine voles, *Microtus pinetorum*, and Ma *et al.* [57] from deer mice, *Peromyscus maniculatus*. In other taxa, DelBarco-Trillo *et al.* [64] detected 74 VOCs in the urine of 12 strepsirrhine primate species; Andersen and Vulpius [54] 55 VOCs in lions, *Panthera leo*, and Soso and Koziel [78] 32 VOCs in Siberian tigers, *Panthera tigris altaica*. The VOCs identified in this study were mainly hydrocarbons, alcohols, aldehydes and ketones, sometimes containing an aromatic group, many of which have been reported in urine of rodents and other mammals, ranging from carnivores to primates (tables 1 and 2; [107]). When inspecting changes in relative peak area for each VOC between the fertile (i.e. P, PE and E stages) and non-fertile phase of oestrous, we found that 3,4-dehydro-exo-brevicomin (in samples analysed on the portable instrument but not in those analysed on the benchtop instrument), butanoic acid, pent-1-ene/cyclopentane, 1,2,3-/1,2,4-trimethylbenzene, heptadecane, dioctyl ether, dodecan-1-ol and 2-ethylhexyl salicylate were reliably present in higher abundance or exclusively in samples collected during the fertile phase. Dehydro-exo-brevicomin (2,3- or 3,4-isomer) is a known semiochemical in both male and female mouse urine [20,27,73,74]. Andreolini *et al.* [20] notably reported a similar increase in the concentration of dehydro-exo-brevicomin in the urine of female mice in oestrus, compared with non-fertile phases. Soini *et al.* [27] found higher levels of dodecan-1-ol in intact male than in castrated male mound-building mice. However, no reference to dodecan-1-ol, nor to the other aforementioned VOCs, was found in the literature regarding female cues of oestrus. Further research, including definitive identification using commercially available chemicals, and the use of bioassays to test the response of male mice to these VOCs [28], will help to draw conclusions about the potential semiochemical role of dehydro-exo-brevicomin, butanoic acid, pent-1-ene/cyclopentane, 1,2,3-/1,2,4-trimethylbenzene, heptadecane, dioctyl ether, dodecan-1-ol and 2-ethylhexyl salicylate in advertising female mouse fertility.

## 5. Conclusion

Vaginal cytology revealed the variable length of oestrous cycles between individual female mice. Portable GC-MS devices like the PerkinElmer Torion® GC-MS are less effective at detecting complex mammalian VOCs when compared with that of traditional benchtop GC-MS systems due to limited compound separation and detection capacities. Optimization of GC settings for urinary compounds such as adjustments to temperature and run time may lead to improved sensitivity. Nonetheless, portable GC-MS systems may have some potential for *in situ* mammalian studies, especially at remote field sites since they allow for the immediate interpretation of results. They could provide researchers with additional flexibility and support following initial analyses of complex samples on a benchtop

GC-MS. Lastly, behavioural assays will help us draw conclusions about the reproductive significance of the VOCs identified here.

Ethics. Experimental procedures were in accordance with the guides approved by the Canadian Council on Animal Care and the University of Calgary Animal Care Committee (no. AC18-0142).

Data accessibility. Datasets and R code are available at Dryad: https://doi.org/10.5061/dryad.nk98sf7t1 [108]. Supplementary figures and tables can be found in the electronic supplementary material [109].

Authors' contributions. A.D.M. and J.T. conceived the study. J.T., L.A.A.M. and G.D. carried out the experiments. O.N. performed chemical analyses; A.C.P. and O.N. carried out chemical interpretations. A.C.P. completed data analyses. J.T and A.C.P. drafted the manuscript, and A.D.M. and O.N. helped in writing the manuscript. All authors gave the final approval for publication.

Competing interests. The authors declare no competing interests.

Funding. J.T. was funded by an undergraduate student research award from the Natural Sciences and Engineering Council of Canada (NSERC). A.C.P. was funded by an NSERC Discovery Accelerator Grant to A.D.M. O.N. was funded by the German Science Foundation (DFG – Deutsche Forschungsgemeinschaft) grant nos. NE2156/1-1 and NE2156/3-1. L.A.A.M. was funded by the Faculty of Graduate Studies (FGS – University of Calgary) and the Alberta Graduate Excellence Scholarship (AGES). A.D.M. was funded by a Discovery Grant (NSERC) and the Canada Research Chairs program.

Acknowledgements. We thank Manfred Ayasse (Ulm University, Germany) for providing laboratory access.

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
