## [Peer Review File · Royal Society Open Science]

Review History

RSOS-210172.R0 (Original submission)

Review form: Reviewer 1

Is the manuscript scientifically sound in its present form?

No

Are the interpretations and conclusions justified by the results?

No

Is the language acceptable?

Yes

Do you have any ethical concerns with this paper?

No

Have you any concerns about statistical analyses in this paper?

No

Recommendation?

Major revision is needed (please make suggestions in comments)

Comments to the Author(s)

The manuscript "Assessing urinary odours across the oestrous cycle in a mouse model using portable and benchtop GC-MS" reports interesting and novel results. However, it cannot be recommended for publication before providing important parameters of GC-MS instruments. In addition, to compare the method based on chromatoprobe with a benchtop TDU-GC-MS with the method based on SPME-GC-MS, authors need to make sure both of them are properly optimized, e.g., for highest sensitivity. Comments:

- 1) Lines 220-230: authors should provide all parameters of the TDU and GC inlet including mode (split or splitless), split ratios, flow rates, liner type.
- 2) Line 238: It is unclear why extraction time 2 minutes was used. It is very low for HS-SPME of liquid samples at atmospheric pressure.
- 3) Line 237: It is unclear why PDMS/DVB fiber was used. DVB/Car/PDMS should be more suitable for extraction of the compounds with a wider range of polarity and volatility.
- 4) In the manuscript, I did not find the volume of the vials used for HS-SPME.
- 5) Line 239: It is unclear why authors conducted the desorption at 10:1 split. For highest responses of analytes, splitless desorption could be used without a problem of tailing peaks (at 1 mL/min flow and 0.75-1 mm i.d. liner). What were the dimensions of the liner and column flow rate?
- 6) Lines 232-249: I did not find the column type and dimensions used with a portable GC-MS.
- 7) Line 369: 38.7 ± 6.5 VOCs (two significant figures in standard deviation should be enough if the first digit is not 1)
- 8) Lines 506-507: only require a pump to be used
- 9) Line 506: TDU is used for thermal desorption, not chemical

Review form: Reviewer 2

Is the manuscript scientifically sound in its present form?

No

Are the interpretations and conclusions justified by the results?

Yes

Is the language acceptable?

Yes

Do you have any ethical concerns with this paper?

No

Have you any concerns about statistical analyses in this paper?

No

Recommendation?

Major revision is needed (please make suggestions in comments)

Comments to the Author(s)

Manuscript RSOS-210172 evaluates odorants of mouse urine, with two main goals: 1) to detect differences in odorants across oestrus stages, and 2) compare results obtained from a benchtop GC-MS to a portable GC-MS device. The study uses vaginal cytology to evaluate oestrus stages. The authors' analysis is thorough and investigates a worthwhile question. I'd like to raise a few issues within the manuscript that would benefit from further attention.

First, the manuscript has major elements of the methods that are unclear. These include: 1) the potential effect of switching methods of vaginal cytology halfway through the study, and 2) a lack of clarity in the sampling procedure and sample size.

Line 140-145 indicates that the authors used saline swabs for (the first?) 13 days of project, but then switched to vaginal lavage, since the swabs yielded "cell densities that were sometimes too low for oestrus stage identification" (l. 141-141). Later in the results, the authors indicate that saline swabs were used for days 1-10 (not for 13 days as indicated in l.144), and lavage for days 11-27. However, they also indicate that data from swabs were discarded (l. 355-356). Were all discarded? If so, why is the sampling period still represented as 27 days (l. 136)? If no data in the manuscript were from these samples, then why is this method included at all? The authors should clarify whether data presented in the manuscript were gathered from two different methods, and if so, provide evidence that they produce equivalent results. Could the lavage yield different types of cells than the swab, for instance by sampling the vaginal cavity more widely or deeply? Is the risk of cervical stimulation and pseudopregnancy (re: l. 139) different (l. 452 notes a problem of prolonged oestrus stages in the finding but doesn't elaborate)?

To compound this problem further, the supplementary material also shows different sampling days from the main text. Table S1 indicates that urine samples from days 1-17 were analyzed, contrasting with either the number and/or labeling of days from the two different timelines in the main text. Lastly, Table S2 appears to count fully through days 1-29 (not 27) in the cells, even though the header suggests only 17 consecutive days were sampled. Overall, it is ambiguous for readers how many days mice were sampled for, and whether or not data from these sample days were collected with two different sampling procedures.

There are also inconsistencies in reporting the analyzed sample size, beyond the number of days sampled. L. 190 indicates that urine collection occurred following vaginal cytology sampling, implying an equal numbers of urine samples (for odorants) and cytology samples (for oestrus classification). If either days 1-10 or 1-13 of the swab technique were discarded (re: l. 144 or l. 356) from the 27 total days, then this should be N=170 (17 days x 10 mice) or N=140 (14 days) samples. However, l. 206 & 232 indicates that only N=90 urine samples were analyzed via GC-MS (N=40 by both the benchtop and portable unit, and an additional N=50 by the portable unit). Likewise Table S1 suggests that only certain days were sampled for each mice, as does l. 557 of the discussion. As far as I can ascertain, there are no details are provided on whether this reduced sample size was due to inconsistent gathering of samples, or whether only a certain subset of samples were chosen for analysis, and why.

This manuscript would also benefit from better situating the context and novelty of its aims within the available literature. Both of the aims of the study, to 1) characterize mouse odorants across estrus cycles, and 2) compare the efficacy of portable and benchtop GC-MS units, have been studied before, in literature cited by the authors. However, the introduction does not provide this important piece of background for readers. Notably, the Andreolini et al. (1987) citation (l. 653) contains data very similar to current study, having both the same aims and methods, in which vaginal cytology is used to analyze mouse oestrus stages in relation to odorant compounds from urine analyzed with GC-MS. Given this, what are the real novel aims of this manuscript beyond this previous work? Is it technological advancement in GC-MS since 1987

that warrants reevaluation of these data? Or was this previous research somehow deficient and/or in need of replication? GC-MS has certainly advanced in the 20+ years since Andreolini et al. (1987), and the current manuscript does report much more detailed findings. Nonetheless, it is important for readers to be aware that this research aim has been previously conducted, and the results of each should be directly compared, particularly since the Andreolini et al. publication DID find differences in odorants between oestrus cycles, while the current manuscript did not. Overall, the introduction leaves readers with the impression that we know little about odorants in mice or how they vary with oestrus. However, the literature presented in the discussion (l. 564-571) suggest that this is not entirely the case.

This is also true for the second aim, comparing portable to benchtop GC-MS devices, albeit to a lesser degree. This comparison has been previously been performed by Kücklich et al. (2017) and Poirier (2019). The current authors do compare their results to these previous studies in the discussion, but do not mention that this comparison (with different device models) has already been done in the introduction, when stating their goals. Again, what is the exact utility here – to compare a different model of portable GC-MS? A different run time? Extraction method? Being more exact about how the current study builds on previous findings would better equip the reader to utilize the information in this manuscript. Overall, the introduction leaves readers with the impression that these aims have not previously been studied, which is not supported by the literature.

Line by line comments

l. 30-31 (+ l. 481-483) This direct comparison implies to readers that the samples are expected to be identical when analyzed on different devices...but is this assumption really true? Nair et al. (2018) has a nice discussion of the idea that of the many different methods available to sample odorants, each is impacted by collection methods, media for preservation (with differing retention biases), and analysis devices (that differ in the detection abilities) that will impact the type of compounds that are retained and detected. In short, while all methods are sampling the same population of compounds, the samples derived from each will inherently differ, based on different collection and storage procedures, and the inherent capacities of the GC-MS devices there are analyzed on. Comparing the samples obtained from two devices and methods used in the manuscript is a worthwhile endeavor. However, several places in the text seem to present this as a direct ‘apples to apples’ comparison – with the implicit assumption that samples obtained should be the same, despite that this does not really portray the nuanced situations of these differing methods.

Nair, J. V., Shanmugam, P. V., Karpe, S. D., Ramakrishnan, U., & Olsson, S. (2018). An optimized protocol for large-scale in situ sampling and analysis of volatile organic compounds. *Ecology and Evolution*, 8(11), 5924–5936.

l. 32 It would help readers to know why these particular compounds are ‘notable’ here.

l. 37 What exactly is meant by ‘subtle’ here? Low abundance of certain compounds? A limited number of compounds that vary between stages? A more exact description would help readers.

l. 73-75 As indicated above, I think it would help readers to more fully describe what is known about mouse urine odorants (re: the literature cited in l. 565-569).

l. 90 Update “in review” or cite thesis, as appropriate.

- l. 104-105 “These data” and also, this should be revised with the above comments. Given that research on this topic has already been done, what further advances in husbandry does the current study provide?
- l. 112 Recommend delete “proof-of-concept” or greatly refine what is meant here. Kücklich et al. (2017) conducted a similar, direct comparison of portable and benchtop GC-MS, seemingly already “proving the concept” demonstrated in the current manuscript. It is unclear that the captive methods here, particularly daily urine samples and vaginal swabs/lavage, would be possible for most wild field studies.
- l. 135-137 As indicated above, it is unclear whether mice were sampled for 27 days, and/or whether 27 days of data are represented in the sample.
- l. 140-145 As indicated above, it needs to be clarified whether the data from the swabs are present in the current manuscript. If so, then evidence that these methods produce equivalent results, and can be appropriately pooled, should be provided.
- l. 161+ I recommend providing a citation for the methods to classify oestrus status.
- l. 169-170 Is this a standardized method, or a new one? There has been much literature on categorizing mouse oestrus, it would be helpful to know if the authors are following established recommendations.
- l. 182-186 Please add acronym definitions for the transitional stages too. If possible, I think a color figure would be more readily interpreted by readers.
- l. 193-194 Please provide the mean, SD and range for both the days until analysis and drops of urine. Additionally, evidence should be provided on the stability of VOC held for this long. Even when frozen, some compounds have been shown to degraded within months.
- l. 201-202 It would helpful to add a regression of the relative peak area and sample volume to the supplemental material. This is also used as a variable in the authors’ analyses, and there is some reason to believe that this measure of abundance may be more impacted by volume than simple presence/absence of compounds.
- l. 197-200 Please add a citation for samples being robust to volume changes, as well as some definition of what is meant by “small difference.” The urine samples in this study are all low volume, but the difference between the smallest volume, 2 drops, and the largest, 40 drops, is a 20-fold increase.
- l. 211 Please provide a range, mean, or estimate for “small volume.” Was this standardized?
- l. 215-218 15+ months at 8°C is quite some time for an analysis of highly volatile odorant compounds. Studies on human urine have indicated that even samples frozen at -80°C should be analyzed within 9 months to prevent degradation (e.g. Esfahani et al., 2016). Evidence should be provided to support the claim that degradation in chromatoprobes does not occur over this timeframe.
- Esfahani, S., Sagar, N. M., Kyrou, I., Mozdiak, E., O’Connell, N., Nwokolo, C., ... & Covington, J. A. (2016). Variation in gas and volatile compound emissions from human urine as it ages, measured by an electronic nose. *Biosensors*, 6(1), 4.

l. 232-233 For the samples measured with both methods, which analysis was done first, and was this standardized? L.195-196 indicates that opening the sample tube generates “loss” of VOCs. Systematically conducting one analysis first or second could create bias due to lost VOCs from the previous opening.

l. 241 For Supp Mat, Table S1, it would be helpful to provide data from the benchtop in the table, to better compare samples analyzed both ways.

l. 245-249 Were blanks and controls also run on the benchtop system? How were compounds found in controls treated in the results?

l. 298 It would be helpful to provide a brief explanation of ‘oestrus cycle’ and why this variable is important in models.

l. 352-357 As indicated above, if all swab data were discarded, then this should probably be deleted. In the least, how these data were treated should be addressed earlier in the methods.

Table 1 & 2. Based on the references provided, these table appear to omit data on humans, despite that this is one of the richest sources of data on mammalian urinary VOCs. I also recommend referencing Charpentier et al. 2012, whose supplemental material has a comprehensive review of compounds used in mammalian communication.

Charpentier, M. J. E., Barthes, N., Proffit, M., Bessièrè, J.-M., & Grison, C. (2012). Critical thinking in the chemical ecology of mammalian communication: Roadmap for future studies. *Functional Ecology*, 26(4), 769-774. <https://doi.org/10.1111/j.1365-2435.2012.01998.x>

l. 421 Please state the “metric.” Is this the Bray-Curtis index?

l. 454-455 Light cycle was controlled in this study (l. 126), so it doesn’t seem like a very pertinent explanatory factor for these findings.

l. 465 Missing stages do not appear to be reported in the results. While mean and SD of stage length is given, there is not a mention of overly prolonged stages in the results, either.

l. 468 Reference #73 is from 1941, and references 25 & 26 are from 2012 and 2013, respectively. Given this age, why were these optimal methods not already incorporated in this study? Presenting these as a way to improve data collection in the future does not seem appropriate.

l. 481-483 This should be expanded on. These instruments have different sampling capacities, and sampling and extraction procedures differed between the two protocols. Since these procedures can impact which compounds are retained, and the devices themselves have different capacities to detect compounds, should these results really be the same? I have no problem with the idea of comparing the results obtained from portable and benchtop models, but I think this text could better portray for readers that there are many reasons that these sampling methods are not able or expected to produce similar results.

l. 504-506 Further information should be provided on the shelf stability of compounds gathered in chromatoprobes.

l. 544-545 Does this capacity differ between the two devices?

l. 557 I believe this is the first mention that samples were not collected every day, and adds to the confusion of the sampling procedure and sample size. There need to be explicit methods given on which days mice were samples, which sample were analyzed and why.

l. 564-571 This body of literature, particularly Andreolini et al. (1987), should be more thoroughly covered in the introduction.

Supplementary Table S1. Adding a data column for the number of VOCs from the benchtop analyzed samples would be helpful. Why were some days not sampled for mice, and why does this differ between mice? Were control and blank compounds removed from these tallies?

Supplementary Table S2. The 'days' headers are confusing. Why are there two overlapping headers for day, and why are days 13-17 both on the first and second line? Why are there 29 days in column headers, but 17 indicated in the caption?

Decision letter (RSOS-210172.R0)

Dear Ms Tang

The Editors assigned to your paper RSOS-210172 "Assessing urinary odours across the oestrous cycle in a mouse model using portable and benchtop GC-MS" have now received comments from reviewers and would like you to revise the paper in accordance with the reviewer comments and any comments from the Editors. Please note this decision does not guarantee eventual acceptance.

Please submit your revised manuscript and required files (see below) no later than 21 days from today's (ie 23-Mar-2021) date. Note: the ScholarOne system will 'lock' if submission of the revision is attempted 21 or more days after the deadline. If you do not think you will be able to meet this deadline please contact the editorial office immediately.

on behalf of Professor Andre Ganswindt (Associate Editor) and Kevin Padian (Subject Editor)
openscience@royalsociety.org

Editor comments to author:

Thank you for your submission. As you see the reviewers had a variety of concerns that they think you can address, but because they are extensive I encourage you to contact our editorial office if you need more time than our standard "major revision" frame allows. Best wishes for your revision.

Reviewer comments to Author:

Reviewer: 1

Comments to the Author(s)

The manuscript "Assessing urinary odours across the oestrous cycle in a mouse model using portable and benchtop GC-MS" reports interesting and novel results. However, it cannot be recommended for publication before providing important parameters of GC-MS instruments. In addition, to compare the method based on chromatoprobe with a benchtop TDU-GC-MS with the method based on SPME-GC-MS, authors need to make sure both of them are properly optimized, e.g., for highest sensitivity. Comments:

- 1) Lines 220-230: authors should provide all parameters of the TDU and GC inlet including mode (split or splitless), split ratios, flow rates, liner type.
- 2) Line 238: It is unclear why extraction time 2 minutes was used. It is very low for HS-SPME of liquid samples at atmospheric pressure.
- 3) Line 237: It is unclear why PDMS/DVB fiber was used. DVB/Car/PDMS should be more suitable for extraction of the compounds with a wider range of polarity and volatility.
- 4) In the manuscript, I did not find the volume of the vials used for HS-SPME.
- 5) Line 239: It is unclear why authors conducted the desorption at 10:1 split. For highest responses of analytes, splitless desorption could be used without a problem of tailing peaks (at 1 mL/min flow and 0.75-1 mm i.d. liner). What were the dimensions of the liner and column flow rate?
- 6) Lines 232-249: I did not find the column type and dimensions used with a portable GC-MS.
- 7) Line 369: 38.7 ± 6.5 VOCs (two significant figures in standard deviation should be enough if the first digit is not 1)
- 8) Lines 506-507: only require a pump to be used
- 9) Line 506: TDU is used for thermal desorption, not chemical

Reviewer: 2

Comments to the Author(s)

Manuscript RSOS-210172 evaluates odorants of mouse urine, with two main goals: 1) to detect differences in odorants across oestrus stages, and 2) compare results obtained from a benchtop GC-MS to a portable GC-MS device. The study uses vaginal cytology to evaluate oestrus stages. The authors' analysis is thorough and investigates a worthwhile question. I'd like to raise a few issues within the manuscript that would benefit from further attention.

First, the manuscript has major elements of the methods that are unclear. These include: 1) the potential effect of switching methods of vaginal cytology halfway through the study, and 2) a lack of clarity in the sampling procedure and sample size.

Line 140-145 indicates that the authors used saline swabs for (the first?) 13 days of project, but then switched to vaginal lavage, since the swabs yielded “cell densities that were sometimes too low for oestrus stage identification” (l. 141-141). Later in the results, the authors indicate that saline swabs were used for days 1-10 (not for 13 days as indicated in l.144), and lavage for days 11-27. However, they also indicate that data from swabs were discarded (l. 355-356). Were all discarded? If so, why is the sampling period still represented as 27 days (l. 136)? If no data in the manuscript were from these samples, then why is this method included at all? The authors should clarify whether data presented in the manuscript were gathered from two different methods, and if so, provide evidence that they produce equivalent results. Could the lavage yield different types of cells than the swab, for instance by sampling the vaginal cavity more widely or deeply? Is the risk of cervical stimulation and pseudopregnancy (re: l. 139) different (l. 452 notes a problem of prolonged oestrus stages in the finding but doesn't elaborate)?

To compound this problem further, the supplementary material also shows different sampling days from the main text. Table S1 indicates that urine samples from days 1-17 were analyzed, contrasting with either the number and/or labeling of days from the two different timelines in the main text. Lastly, Table S2 appears to count fully through days 1-29 (not 27) in the cells, even though and the header suggests only 17 consecutive days were sampled. Overall, it is ambiguous for readers how many days mice were sampled for, and whether or not data from these sample days were collected with two different sampling procedures.

There are also inconsistencies in reporting the analyzed sample size, beyond the number of days sampled. L. 190 indicates that urine collection occurred following vaginal cytology sampling, implying an equal numbers of urine samples (for odorants) and cytology samples (for oestrus classification). If either days 1-10 or 1-13 of the swab technique were discarded (re: l. 144 or l. 356) from the 27 total days, then this should be N=170 (17 days x 10 mice) or N=140 (14 days) samples. However, l. 206 & 232 indicates that only N=90 urine samples were analyzed via GC-MS (N=40 by both the benchtop and portable unit, and an additional N=50 by the portable unit). Likewise Table S1 suggests that only certain days were sampled for each mice, as does l. 557 of the discussion. As far as I can ascertain, there are no details are provided on whether this reduced sample size was due to inconsistent gathering of samples, or whether only a certain subset of samples were chosen for analysis, and why.

This manuscript would also benefit from better situating the context and novelty of its aims within the available literature. Both of the aims of the study, to 1) characterize mouse odorants across estrus cycles, and 2) compare the efficacy of portable and benchtop GC-MS units, have been studied before, in literature cited by the authors. However, the introduction does not provide this important piece of background for readers. Notably, the Andreolini et al. (1987) citation (l. 653) contains data very similar to current study, having both the same aims and methods, in which vaginal cytology is used to analyze mouse oestrus stages in relation to odorant compounds from urine analyzed with GC-MS. Given this, what are the real novel aims of this manuscript beyond this previous work? Is it technological advancement in GC-MS since 1987 that warrants reevaluation of these data? Or was this previous research somehow deficient and/or in need of replication? GC-MS has certainly advanced in the 20+ years since Andreolini et al. (1987), and the current manuscript does report much more detailed findings. Nonetheless, it is important for readers to be aware that this research aim has been previously conducted, and the results of each should be directly compared, particularly since the Andreolini et al. publication DID find differences in odorants between oestrus cycles, while the current manuscript did not. Overall, the introduction leaves readers with the impression that we know little about odorants in

mice or how they vary with oestrus. However, the literature presented in the discussion (l. 564-571) suggest that this is not entirely the case.

This is also true for the second aim, comparing portable to benchtop GC-MS devices, albeit to a lesser degree. This comparison has been previously been performed by Kücklich et al. (2017) and Poirier (2019). The current authors do compare their results to these previous studies in the discussion, but do not mention that this comparison (with different device models) has already been done in the introduction, when stating their goals. Again, what is the exact utility here – to compare a different model of portable GC-MS? A different run time? Extraction method? Being more exact about how the current study builds on previous findings would better equip the reader to utilize the information in this manuscript. Overall, the introduction leaves readers with the impression that these aims have not previously been studied, which is not supported by the literature.

Line by line comments

l. 30-31 (+ l. 481-483) This direct comparison implies to readers that the samples are expected to be identical when analyzed on different devices...but is this assumption really true? Nair et al. (2018) has a nice discussion of the idea that of the many different methods available to sample odorants, each is impacted by collection methods, media for preservation (with differing retention biases), and analysis devices (that differ in the detection abilities) that will impact the type of compounds that are retained and detected. In short, while all methods are sampling the same population of compounds, the samples derived from each will inherently differ, based on different collection and storage procedures, and the inherent capacities of the GC-MS devices there are analyzed on. Comparing the samples obtained from two devices and methods used in the manuscript is a worthwhile endeavor. However, several places in the text seem to present this as a direct ‘apples to apples’ comparison – with the implicit assumption that samples obtained should be the same, despite that this does not really portray the nuanced situations of these differing methods.

Nair, J. V., Shanmugam, P. V., Karpe, S. D., Ramakrishnan, U., & Olsson, S. (2018). An optimized protocol for large-scale in situ sampling and analysis of volatile organic compounds. *Ecology and Evolution*, 8(11), 5924–5936.

l. 32 It would help readers to know why these particular compounds are ‘notable’ here.

l. 37 What exactly is meant by ‘subtle’ here? Low abundance of certain compounds? A limited number of compounds that vary between stages? A more exact description would help readers.

l. 73-75 As indicated above, I think it would help readers to more fully describe what is known about mouse urine odorants (re: the literature cited in l. 565-569).

l. 90 Update “in review” or cite thesis, as appropriate.

l. 104-105 “These data” and also, this should be revised with the above comments. Given that research on this topic has already been done, what further advances in husbandry does the current study provide?

l. 112 Recommend delete “proof-of-concept” or greatly refine what is meant here. Kücklich et al. (2017) conducted a similar, direct comparison of portable and benchtop GC-MS, seemingly already “proving the concept” demonstrated in the current manuscript. It is unclear that the captive methods here, particularly daily urine samples and vaginal swabs/lavage, would be possible for most wild field studies.

- l. 135-137 As indicated above, it is unclear whether mice were sampled for 27 days, and/or whether 27 days of data are represented in the sample.
- l. 140-145 As indicated above, it needs to be clarified whether the data from the swabs are present in the current manuscript. If so, then evidence that these methods produce equivalent results, and can be appropriately pooled, should be provided.
- l. 161+ I recommend providing a citation for the methods to classify oestrus status.
- l. 169-170 Is this a standardized method, or a new one? There has been much literature on categorizing mouse oestrus, it would be helpful to know if the authors are following established recommendations.
- l. 182-186 Please add acronym definitions for the transitional stages too. If possible, I think a color figure would be more readily interpreted by readers.
- l. 193-194 Please provide the mean, SD and range for both the days until analysis and drops of urine. Additionally, evidence should be provided on the stability of VOC held for this long. Even when frozen, some compounds have been shown to degraded within months.
- l. 201-202 It would helpful to add a regression of the relative peak area and sample volume to the supplemental material. This is also used as a variable in the authors' analyses, and there is some reason to believe that this measure of abundance may be more impacted by volume than simple presence/absence of compounds.
- l. 197-200 Please add a citation for samples being robust to volume changes, as well as some definition of what is meant by "small difference." The urine samples in this study are all low volume, but the difference between the smallest volume, 2 drops, and the largest, 40 drops, is a 20-fold increase.
- l. 211 Please provide a range, mean, or estimate for "small volume." Was this standardized?
- l. 215-218 15+ months at 8°C is quite some time for an analysis of highly volatile odorant compounds. Studies on human urine have indicated that even samples frozen at -80°C should be analyzed within 9 months to prevent degradation (e.g. Esfahani et al., 2016). Evidence should be provided to support the claim that degradation in chromatoprobes does not occur over this timeframe.
- Esfahani, S., Sagar, N. M., Kyrou, I., Mozdiak, E., O'Connell, N., Nwokolo, C., ... & Covington, J. A. (2016). Variation in gas and volatile compound emissions from human urine as it ages, measured by an electronic nose. *Biosensors*, 6(1), 4.
- l. 232-233 For the samples measured with both methods, which analysis was done first, and was this standardized? L.195-196 indicates that opening the sample tube generates "loss" of VOCs. Systematically conducting one analysis first or second could create bias due to lost VOCs from the previous opening.
- l. 241 For Supp Mat, Table S1, it would be helpful to provide data from the benchtop in the table, to better compare samples analyzed both ways.
- l. 245-249 Were blanks and controls also run on the benchtop system? How were compounds found in controls treated in the results?

l. 298 It would be helpful to provide a brief explanation of 'oestrus cycle' and why this variable is important in models.

l. 352-357 As indicated above, if all swab data were discarded, then this should probably be deleted. In the least, how these data were treated should be addressed earlier in the methods.

Table 1 & 2. Based on the references provided, these table appear to omit data on humans, despite that this is one of the richest sources of data on mammalian urinary VOCs. I also recommend referencing Charpentier et al. 2012, whose supplemental material has a comprehensive review of compounds used in mammalian communication.

Charpentier, M. J. E., Barthes, N., Proffit, M., Bessièrè, J.-M., & Grison, C. (2012). Critical thinking in the chemical ecology of mammalian communication: Roadmap for future studies. *Functional Ecology*, 26(4), 769-774. <https://doi.org/10.1111/j.1365-2435.2012.01998.x>

l. 421 Please state the "metric." Is this the Bray-Curtis index?

l. 454-455 Light cycle was controlled in this study (l. 126), so it doesn't seem like a very pertinent explanatory factor for these findings.

l. 465 Missing stages do not appear to be reported in the results. While mean and SD of stage length is given, there is not a mention of overly prolonged stages in the results, either.

l. 468 Reference #73 is from 1941, and references 25 & 26 are from 2012 and 2013, respectively. Given this age, why were these optimal methods not already incorporated in this study? Presenting these as a way to improve data collection in the future does not seem appropriate.

l. 481-483 This should be expanded on. These instruments have different sampling capacities, and sampling and extraction procedures differed between the two protocols. Since these procedures can impact which compounds are retained, and the devices themselves have different capacities to detect compounds, should these results really be the same? I have no problem with the idea of comparing the results obtained from portable and benchtop models, but I think this text could better portray for readers that there are many reasons that these sampling methods are not able or expected to produce similar results.

l. 504-506 Further information should be provided on the shelf stability of compounds gathered in chromatoprobes.

l. 544-545 Does this capacity differ between the two devices?

l. 557 I believe this is the first mention that samples were not collected every day, and adds to the confusion of the sampling procedure and sample size. There need to be explicit methods given on which days mice were samples, which sample were analyzed and why.

l. 564-571 This body of literature, particularly Andreolini et al. (1987), should be more thoroughly covered in the introduction.

Supplementary Table S1. Adding a data column for the number of VOCs from the benchtop analyzed samples would be helpful. Why were some days not sampled for mice, and why does this differ between mice? Were control and blank compounds removed from these tallies?

Supplementary Table S2. The 'days' headers are confusing. Why are there two overlapping headers for day, and why are days 13-17 both on the first and second line? Why are there 29 days in column headers, but 17 indicated in the caption?

===PREPARING YOUR MANUSCRIPT===

===PREPARING YOUR REVISION IN SCHOLARONE===

Author's Response to Decision Letter for (RSOS-210172.R0)

See Appendix A.

RSOS-210172.R1 (Revision)

Review form: Reviewer 1

Is the manuscript scientifically sound in its present form?

Yes

Are the interpretations and conclusions justified by the results?

Yes

Is the language acceptable?

Yes

Do you have any ethical concerns with this paper?

No

Have you any concerns about statistical analyses in this paper?

No

Recommendation?

Accept with minor revision (please list in comments)

Comments to the Author(s)

The manuscript has improved, but few minor issues still have to be resolved:

- 1) Page 8, Line 43: do authors mean CIS4 instead of CIC 4C?
- 2) Page 8, Line 45: "Samples were introduced splitless to the thermal desorption unit (TDU) at 30°C" - maybe samples were desorbed in the thermal desorption unit in splitless mode? What was the flow rate in TDU during desorption?
- 3) Page 8, Lines 53-55: "We used helium at a flow of 1 mL/min as carrier gas." - what was the flow rate of He in the liner during desorption of analytes from liner to the column? Was the CIS in splitless mode during this step?
- 4) Page 9, Line 16: better used commercial fiber title: PDMS/DVB instead of DVB/PDMS

Review form: Reviewer 2

Is the manuscript scientifically sound in its present form?

No

Are the interpretations and conclusions justified by the results?

Yes

Is the language acceptable?

Yes

Do you have any ethical concerns with this paper?

No

Have you any concerns about statistical analyses in this paper?

No

Recommendation?

Major revision is needed (please make suggestions in comments)

Comments to the Author(s)

I appreciate the opportunity to once again review manuscript RSOS-210172. The authors have made a number of revisions that have improved the quality of the manuscript. This includes refining the background information given to readers, clarifying the novelty of the study, and

reporting the sample size and sampling days in a more straightforward manner. However, there are still a few remaining items that I do not believe have been satisfactorily addressed.

First, the data collection methods for vaginal cytology were switched part way through the study, but the manuscript does not provide any evidence that the two methods (swab and lavage) produce equivalent results. While this version better clarifies for readers what method was used when, there is still not assurance that the methods produced consistent results. I recommend the authors either: 1) cite references to support the equivalency of these methods, 2) provide any possible supplemental data from the current study that demonstrates equivalency, or 3) delete the two days of data collected by swabs, so that all data are from the lavage method.

Second, the samples in this study were stored for much longer, and at a warmer temperature, than recommended by references cited by the authors. Yet, several points in the manuscript's text suggest that this storage period is not problematic, or downplay the time frame (pg. 7 l. 45-50, pg. 8 l. 35-37, pg. 17 l. 46-51, pg. 20 l. 19-23). This assertion is inconsistent with the literature. Saude & Sykes (2007), which is cited by the manuscript, shows severe degradation of VOCs after four weeks when stored at 4°C, and consistent (although much less severe) degradation after 4 weeks when stored at -80°C. By contrast, the current manuscript stored urine at -20°C for 14-29 weeks (100-205 days: pg. 7 l. 46 – and possibly an additional year in chromatoprobes; see below comments). Likewise, and as pointed out in my previous review, Esfahani et al. (2016) recommended that samples be frozen at -80°C and analyzed within 9 months. Overall, the storage of odorant samples does not appear to be in line with current guidelines. Based on the previous literature, it seems likely this study experienced degradation of samples. This needs to be better acknowledged for readers, the text should be modified to more accurately portray the literature on VOC storage degradation, and appropriate context for the storage time frames used in this study, relative to the published literature, needs to be provided.

The manuscript also contains similar statements about storage of chromatoprobes which is equally problematic. Pg. 8 l. 35-37 indicates that chromatoprobes were stored for 450 days prior to analysis. The manuscript text indicates that these have stable retention during storage (pg. 8 l. 35-37, pg. 17 l. 46-51, pg. 20 l. 19-23), however no references, or data, are given to support this assertion. Instead, the authors cite “unpublished data” as support, without providing readers with any details on this information. If the authors have data that these chromatoprobes can prevent VOC degradation (and/or retention biases) for over 450 days, then I strongly encourage them to publish this as a separate paper, as it is a significant finding. However, given the long-standing literature showing degradation of VOCs in a variety of media, and the extended time frame of storage in this study (129 days on average as urine, plus 450 days in a chromatoprobe), readers need to be presented with solid evidence to support the assertion that degradation is within acceptable limits. Vague references to unpublished data are not sufficient.

Lastly, while the authors have clarified their sampling procedure in this revision, there is still no explanation given for how the subset of samples for odor analysis were selected. While there were clearly constraints on analyzing all samples collected, supplement table S1 suggests that selection of samples was not random. The selection procedure should be described for readers, with any pertinent notes on avoiding selection biases (conscious or unconscious).

Detailed comments (Please note that page numbers refer to labeled page numbers on the tracked changes version of the manuscript, not the proof PDF's page numbers).

Pg. 7 l. 45 & Pg. 8 l. 35 The total storage times should be better clarified for each method. I assume this means that all urine was stored for an average of 129 days, while the benchtop samples were stored for an additional 450 days in a chromatoprobe. This would yield a total

storage time of around 1.5 years (which is sizeable). Were the samples analyzed via portable GC-MS measured after the timeframe provided in pg 7 l. 45?

Pg 7 l. 46-49 While this statement acknowledges that some degradation may occur, it does not seem to accurately portray this study (noted above), which recommended that urine be stored at -80°C (a frequent recommendation in VOC research which was not taken in the current study), and still found degradation at that colder temperature, over just a fraction of the time that samples were stored for in the current study.

Pg 7 l. 59 – pg. 8 l. 5 The argument here seems to be that standardizing sample volume doesn't matter because it is really urine concentration that dictates the number of VOC detected and peak area. However, the study does not appear to measure or standardize urine concentration either, so it is unclear how helpful or effective this comment is. This addition seems to undercut the reason for providing the information in Supplemental Figure S1.

Pg. 8 l. 14-16 Details on the selection procedure for the subset of odorant samples that were analyzed should be provided.

Pg 8 l. 35-36 This appears to contradict the literature on the storage of VOCs. Evidence that VOCs can survive 450 days in a chromatoprobe (on top of 129+ days frozen?) needs to be provided for readers. Citing past personal experience, with no data or peer-reviewed literature to support it, is not an acceptable level of rigor for publication.

Pg. 9 l. 3-6 Details on the subset selection procedure should be provided here as well. It is not clear what “representative” refers to (Oestrus cycle? Volume? Odors?), particularly since this sample size overlaps with, and is larger than, the subset of samples analyzed for odors via benchtop.

Pg. 16, l. 16 Data on the “efficiency” of this method are not reported in the results; it is unclear what result this refers to. Also, the study switched collection methods part way through, so describing this method as “consistent” does not seem accurate.

Pg. 17 l. 46-51 No references are provided to support the assertion that chromatoprobes can preserve VOCs for over three years. If the authors have these data, I strongly encourage them to publish. Given the well documented degradation of VOCs even in ideal conditions and over shorter time frames, a vague reference to unpublished data is not particularly convincing. Likewise referring to this study's storage period of over a year as “short” does not seem to be an accurate descriptor.

Pg. 20 l. 6-7 It is unclear what data this statement (“complex biological nature of urine...”) is referring to. Wasn't the GC-MS method the challenging issue, rather than the complexity of urine? Or is this referring to the discussion of MUP interactions? Given that MUPS and chemical interactions were not measured, it is unclear if this sentence should be included in the conclusions or should be rephrased.

Pg. 20 l. 21-23 As above, evidence or published studies need to be given to support the statement that this method produces stable retention of VOCs. Data on this are not provided in the manuscript, nor are any references provided that support this assertion. This conclusion also seems tangential to the primary study goals to compare benchtop to portable GC-MS devices and characterize VOC changes over the mouse oestrus cycle.

Decision letter (RSOS-210172.R1)

Dear Ms Tang

The Editors assigned to your paper RSOS-210172.R1 "Assessing urinary odours across the oestrous cycle in a mouse model using portable and benchtop GC-MS" have now received comments from reviewers and would like you to revise the paper in accordance with the reviewer comments and any comments from the Editors. Please note this decision does not guarantee eventual acceptance.

Please submit your revised manuscript and required files (see below) no later than 21 days from today's (ie 09-Jun-2021) date. Note: the ScholarOne system will 'lock' if submission of the revision is attempted 21 or more days after the deadline. If you do not think you will be able to meet this deadline please contact the editorial office immediately.

on behalf of Professor Andre Ganswindt (Associate Editor) and Kevin Padian (Subject Editor)
openscience@royalsociety.org

Subject Editor Comments to Author (Professor Kevin Padian):
Comments to the Author:

Thanks for your revision. The concerns that the second reviewer raises about the preservation of the sample, which appear to differ from usual standards, and how that may have affected your results, are serious ones that in the reviewer's judgment have not been adequately addressed previously. Please re-assess and rewrite to take these into consideration, and address them in a

separate document upon resubmission. We regret that we will not be able to send your MS out for another round of review, but this reviewer may be asked to look at it again. Best wishes for your revisions and if you need more time please let the editorial office know.

Reviewer comments to Author:

Reviewer: 1

Comments to the Author(s)

The manuscript has improved, but few minor issues still have to be resolved:

- 1) Page 8, Line 43: do authors mean CIS4 instead of CIC 4C?
- 2) Page 8, Line 45: "Samples were introduced splitless to the thermal desorption unit (TDU) at 30°C" - maybe samples were desorbed in the thermal desorption unit in splitless mode? What was the flow rate in TDU during desorption?
- 3) Page 8, Lines 53-55: "We used helium at a flow of 1 mL/min as carrier gas." - what was the flow rate of He in the liner during desorption of analytes from liner to the column? Was the CIS in splitless mode during this step?
- 4) Page 9, Line 16: better used commercial fiber title: PDMS/DVB instead of DVB/PDMS

Reviewer: 2

Comments to the Author(s)

I appreciate the opportunity to once again review manuscript RSOS-210172. The authors have made a number of revisions that have improved the quality of the manuscript. This includes refining the background information given to readers, clarifying the novelty of the study, and reporting the sample size and sampling days in a more straightforward manner. However, there are still a few remaining items that I do not believe have been satisfactorily addressed.

First, the data collection methods for vaginal cytology were switched part way through the study, but the manuscript does not provide any evidence that the two methods (swab and lavage) produce equivalent results. While this version better clarifies for readers what method was used when, there is still not assurance that the methods produced consistent results. I recommend the authors either: 1) cite references to support the equivalency of these methods, 2) provide any possible supplemental data from the current study that demonstrates equivalency, or 3) delete the two days of data collected by swabs, so that all data are from the lavage method.

Second, the samples in this study were stored for much longer, and at a warmer temperature, than recommended by references cited by the authors. Yet, several points in the manuscript's text suggest that this storage period is not problematic, or downplay the time frame (pg. 7 l. 45-50, pg. 8 l. 35-37, pg. 17 l. 46-51, pg. 20 l. 19-23). This assertion is inconsistent with the literature. Saude & Sykes (2007), which is cited by the manuscript, shows severe degradation of VOCs after four weeks when stored at 4°C, and consistent (although much less severe) degradation after 4 weeks when stored at -80°C. By contrast, the current manuscript stored urine at -20°C for 14-29 weeks (100-205 days: pg. 7 l. 46 – and possibly an additional year in chromatoprobes; see below comments). Likewise, and as pointed out in my previous review, Esfahani et al. (2016) recommended that samples be frozen at -80°C and analyzed within 9 months. Overall, the storage of odorant samples does not appear to be in line with current guidelines. Based on the previous literature, it seems likely this study experienced degradation of samples. This needs to be better acknowledged for readers, the text should be modified to more accurately portray the literature on VOC storage degradation, and appropriate context for the storage times frames used in this study, relative to the published literature, needs to be provided.

The manuscript also contains similar statements about storage of chromatoprobes which is equally problematic. Pg. 8 l. 35-37 indicates that chromatoprobes were stored for 450 days prior to analysis. The manuscript text indicates that these have stable retention during storage (pg. 8 l. 35-

37, pg. 17 l. 46-51, pg. 20 l. 19-23), however no references, or data, are given to support this assertion. Instead, the authors cite “unpublished data” as support, without providing readers with any details on in this information. If the authors have data that these chromatoprobes can prevent VOC degradation (and/or retention biases) for over 450 days, then I strongly encourage them to publish this as a separate paper, as it is a significant finding. However, given the long-standing literature showing degradation of VOCs in a variety of media, and the extended time frame of storage in this study (129 days on average as urine, plus 450 days in a chromatoprobe), readers need to be presented with solid evidence to support the assertion that degradation is within acceptable limits. Vague references to unpublished data are not sufficient.

Lastly, while the authors have clarified their sampling procedure in this revision, there is still no explanation given for how the subset of samples for odor analysis were selected. While there were clearly constraints on analyzing all samples collected, supplement table S1 suggests that selection of samples was not random. The selection procedure should be described for readers, with any pertinent notes on avoiding selection biases (conscious or unconscious).

Detailed comments (Please note that page numbers refer to labeled page numbers on the tracked changes version of the manuscript, not the proof PDF’s page numbers).

Pg. 7 l. 45 & Pg. 8 l. 35 The total storage times should be better clarified for each method. I assume this means that all urine was stored for an average of 129 days, while the benchtop samples were stored for an additional 450 days in a chromatoprobe. This would yield a total storage time of around 1.5 years (which is sizeable). Were the samples analyzed via portable GC-MS measured after the timeframe provided in pg 7 l. 45?

Pg 7 l. 46-49 While this statement acknowledges that some degradation may occur, it does not seem to accurately portray this study (noted above), which recommended that urine be stored at -80°C (a frequent recommendation in VOC research which was not taken in the current study), and still found degradation at that colder temperature, over just a fraction of the time that samples were stored for in the current study.

Pg 7 l. 59 – pg. 8 l. 5 The argument here seems to be that standardizing sample volume doesn’t matter because it is really urine concentration that dictates the number of VOC detected and peak area. However, the study does not appear to measure or standardize urine concentration either, so it is unclear how helpful or effective this comment is. This addition seems to undercut the reason for providing the information in Supplemental Figure S1.

Pg. 8 l. 14-16 Details on the selection procedure for the subset of odorant samples that were analyzed should be provided.

Pg 8 l. 35-36 This appears to contradict the literature on the storage of VOCs. Evidence that VOCs can survive 450 days in a chromatoprobe (on top of 129+ days frozen?) needs to be provided for readers. Citing past personal experience, with no data or peer-reviewed literature to support it, is not an acceptable level of rigor for publication.

Pg. 9 l. 3-6 Details on the subset selection procedure should be provided here as well. It is not clear what “representative” refers to (Oestrus cycle? Volume? Odors?), particularly since this sample size overlaps with, and is larger than, the subset of samples analyzed for odors via benchtop.

Pg. 16, l. 16 Data on the “efficiency” of this method are not reported in the results; it is unclear what result this refer to. Also, the study switched collection methods part way through, so describing this method as “consistent” does not seem accurate.

Pg. 17 l. 46-51 No references are provided to support the assertion that chromatoprobes can preserve VOCs for over three years. If the authors have these data, I strongly encourage them to publish. Given the well documented degradation of VOCs even in ideal conditions and over shorter time frames, a vague reference to unpublished data is not particularly convincing. Likewise referring to this study's storage period of over a year as "short" does not seem to be an accurate descriptor.

Pg. 20 l. 6-7 It is unclear what data this statement ("complex biological nature of urine...") is referring to. Wasn't the GC-MS method the challenging issue, rather than the complexity of urine? Or is this referring to the discussion of MUP interactions? Given that MUPS and chemical interactions were not measured, it is unclear if this sentence should be included in the conclusions or should be rephrased.

Pg. 20 l. 21-23 As above, evidence or published studies need to be given to support the statement that this method produces stable retention of VOCs. Data on this are not provided in the manuscript, nor are any references provided that support this assertion. This conclusion also seems tangential to the primary study goals to compare benchtop to portable GC-MS devices and characterize VOC changes over the mouse oestrus cycle.

===PREPARING YOUR MANUSCRIPT===

===PREPARING YOUR REVISION IN SCHOLARONE===

Author's Response to Decision Letter for (RSOS-210172.R1)

See Appendix B.

RSOS-210172.R2 (Revision)

Review form: Reviewer 1

Is the manuscript scientifically sound in its present form?

Yes

Are the interpretations and conclusions justified by the results?

Yes

Is the language acceptable?

Yes

Do you have any ethical concerns with this paper?

No

Have you any concerns about statistical analyses in this paper?

No

Recommendation?

Accept as is

Comments to the Author(s)

The manuscript can now be recommended for acceptance and publication

Review form: Reviewer 2

Is the manuscript scientifically sound in its present form?

Yes

Are the interpretations and conclusions justified by the results?

Yes

Is the language acceptable?

Yes

Do you have any ethical concerns with this paper?

Yes

Have you any concerns about statistical analyses in this paper?

Yes

Recommendation?

Accept as is

Comments to the Author(s)

The authors have addressed my remaining concerns. While the storage period is still beyond the recommended amount of time, the ms at least more clearly discloses this fact to readers.

Decision letter (RSOS-210172.R2)

Dear Ms Tang,

It is a pleasure to accept your manuscript entitled "Assessing urinary odours across the oestrous cycle in a mouse model using portable and benchtop GC-MS" in its current form for publication in Royal Society Open Science. The comments of the reviewer(s) who reviewed your manuscript are included at the foot of this letter.

on behalf of Professor Andre Ganswindt (Associate Editor) and Kevin Padian (Subject Editor)
openscience@royalsociety.org

Reviewer comments to Author:

Reviewer: 2

Comments to the Author(s)

The authors have addressed my remaining concerns. While the storage period is still beyond the recommended amount of time, the ms at least more clearly discloses this fact to readers.

Reviewer: 1

Comments to the Author(s)

The manuscript can now be recommended for acceptance and publication

Appendix A

Response to the Editors' and Reviewers' comments

Thank you for your submission. As you see the reviewers had a variety of concerns that they think you can address, but because they are extensive I encourage you to contact our editorial office if you need more time than our standard "major revision" frame allows. Best wishes for your revision.

Dear Profs. Ganswindt and Padian,

Thank you very much for your interest in our work, and for giving us the opportunity to improve our manuscript. We have addressed the points raised by the Reviewers in our revised manuscript, as detailed below. We are grateful for the extensive feedback and believe our paper to be much improved. We hope it is now found suitable for publication at Royal Society Open Science.

Best regards,

Jia Tang and Alice Poirier, on behalf of all co-authors

Reviewer comments to Author:

Reviewer: 1

Comments to the Author(s)

The manuscript "Assessing urinary odours across the oestrous cycle in a mouse model using portable and benchtop GC-MS" reports interesting and novel results. However, it cannot be recommended for publication before providing important parameters of GC-MS instruments. In addition, to compare the method based on chromatoprobe with a benchtop TDU-GC-MS with the method based on SPME-GC-MS, authors need to make sure both of them are properly optimized, e.g., for highest sensitivity.

We are grateful for the Reviewer's interest in our manuscript. We have addressed the Reviewer's concerns about the choice of instrument parameters in our revised manuscript, which we detail below. We further address and explain limitations in the optimization of our instruments in the Methods (lines 244-250, 263-270 & 276-280) and Discussion (lines 537-539, 547-549 & 568-573).

Comments:

1) Lines 220-230: authors should provide all parameters of the TDU and GC inlet including mode (split or splitless), split ratios, flow rates, liner type.

We have added details on the parameters used: "Samples were introduced splitless to the thermal desorption unit (TDU) at 30°C. [...] The liner (Gerstel 6817-U glass liner filled with

silanized glass wool) was cooled to -100°C using liquid nitrogen. [...] We used helium at a flow of 1 mL/min as carrier gas.” (lines 244-250).

2) Line 238: It is unclear why extraction time 2 minutes was used. It is very low for HS-SPME of liquid samples at atmospheric pressure.

We followed a 2 minute recommendation made by PerkinElmer, the manufacturer of the Torion GC-MS and the Custodion SPME fibre used in our study. This is within the range used in reports published by PerkinElmer; extraction duration using this technique ranged from 15 sec to 5 min (PerkinElmer 2019). We also experimented and found that in our trials, lengthening the extraction time beyond the times reported here did not increase the number of compounds detected. Increasing the injection temperature had a negative effect on the SPME fibre, causing burning.

We have clarified our Methods: “Temperature and runtimes were based on advice from technical experts at PerkinElmer, and consistent with product recommendations [44]. We experimented and found that in our trials, lengthening the extraction time beyond the times reported here did not increase the number of compounds detected. Increasing the injection temperature had a negative effect on the SPME fibre, causing burning.” (lines 276-280).

We have also added further acknowledgement of the limitations of our methods, and recommendations for future use, in our Discussion section 4.2: “Our study also presented the first use of the Torion® portable GC-MS for the analysis of urinary VOCs. The efficiency of this device appeared limited, yet several methodological aspects may be improved to ensure better results in the future. These include optimizing VOC extraction and detection procedure through the choice of SPME fiber coating, injection parameters, GC column temperature cycle and MS detector parameters. Despite limitations, this instrument or similar devices may still have utility for *in situ* chemical analyses.” (lines 568-573).

PerkinElmer. (2019). *Sample Analysis Onsite with Torion T-9 Portable GC/MS Application Compendium*.

3) Line 237: It is unclear why PDMS/DVB fiber was used. DVB/Car/PDMS should be more suitable for extraction of the compounds with a wider range of polarity and volatility.

Our rationale was to use a very commonly used VOC trap system and to focus our comparison on large differences between portable and benchtop systems. While we agree with the Reviewer’s comment in principle, we believe it is not trivial to decide which absorbent mix is better. Sometimes, fewer types of absorbents give better results (cf. Lorenzo 2014). A thorough optimization of our extraction method would have potentially required years of adjustments to the instrument methods and protocols, which was not feasible for the present study.

We have added further justification for the choice of SPME fibre to our Methods: “DVB/PDMS is a generalist combination of absorbents designed to trap a large range of VOCs.” (lines 262-263).

Lorenzo, J. M. (2014). Influence of the type of fiber coating and extraction time on foal dry-cured loin volatile compounds extracted by solid-phase microextraction (SPME). *Meat Science*, 96(1), 179–186.

4) In the manuscript, I did not find the volume of the vials used for HS-SPME.

We apologize for this omission and have added this information to the manuscript, as well as further details about the vials used: “We collected urine in 4 mL glass screw-top vials fitted with a polytetrafluoroethylene/silicone septum (Supelco) following the collection of vaginal smears.” (lines 204-205).

5) Line 239: It is unclear why authors conducted the desorption at 10:1 split. For highest responses of analytes, splitless desorption could be used without a problem of tailing peaks (at 1 mL/min flow and 0.75-1 mm i.d. liner). What were the dimensions of the liner and column flow rate?

We agree with the Reviewer’s comment that had the samples been injected on splitless mode, we might have detected a greater number of VOCs in the samples, and the signal for all detected peaks would have been greater. Our decision was based on the recommendations given by the Torion manufacturer, which we have now made clearer in the manuscript: “, as recommended by the manufacturer [44]” (line 269).

To briefly evaluate this choice, we conducted chemical analyses of VOCs from commercially purchased bananas using the same portable instrument, which showed no significant improvement in the number of peaks detected or their abundance when using splitless mode (0 to 2 s) than when using 1:10 split at injection (Poirier, unpublished results). However, we now raise this as an area for future consideration in the Discussion (lines 570-573).

Overall, the different GC and MS components were specific to the miniaturized feature of the Torion GC-MS. Unlike with benchtop instruments, it was not possible to adapt the choice of individual GC and MS components to best fit our samples. Instead, the Torion was designed for fast and versatile chemical analyses in a range of fields, from chemical warfare to the food and drug industries.

We have added details on the Torion GC and MS components in the Methods: “The SPME fibre was then desorbed into the injection port of the Torion® GC, fitted with a deactivated 0.048 in i.d. liner, at 270°C for 5 s. The Torion® GC system was fitted with a custom small diameter, low polarity MXT-5 low thermal mass capillary column (5 m x 0.1 mm x 0.4 µm) bundled with electrical resistive heating, allowing for extremely rapid heating and cooling speeds [43]. We used helium at a flow of 0.2 mL/min as carrier gas (disposable cartridges, PerkinElmer). Split mode was applied at 10:1 at 0 s and then 50:1 at 10 s, as recommended by the manufacturer [44]. Mass separation was performed by a toroidal ion trap MS on electron ionization mode at 70 eV [43], scanning between 41-500 Da.” (lines 263-270).

6) Lines 232-249: I did not find the column type and dimensions used with a portable GC-MS.

We have added details on the Torion GC column: “The Torion® GC system was fitted with a custom small diameter, low polarity MXT-5 low thermal mass capillary column (5 m x 0.1 mm x 0.4 µm) bundled with electrical resistive heating, allowing for rapid heating and cooling speeds [43].” (lines 264-267).

7) Line 369: 38.7 ± 6.5 VOCs (two significant figures in standard deviation should be enough if the first digit is not 1)

We have modified all result values to show only one digit numbers (lines 411-415).

8) Lines 506-507: only require a pump to be used

We have corrected the sentence (lines 557).

9) Line 506: TDU is used for thermal desorption, not chemical

We have replaced “chemical desorption” with “thermal desorption” in the sentence (line 557).

Reviewer: 2

Comments to the Author(s)

Manuscript RSOS-210172 evaluates odorants of mouse urine, with two main goals: 1) to detect differences in odorants across oestrus stages, and 2) compare results obtained from a benchtop GC-MS to a portable GC-MS device. The study uses vaginal cytology to evaluate oestrus stages. The authors’ analysis is thorough and investigates a worthwhile question. I’d like to raise a few issues within the manuscript that would benefit from further attention.

First, the manuscript has major elements of the methods that are unclear. These include: 1) the potential effect of switching methods of vaginal cytology halfway through the study, and 2) a lack of clarity in the sampling procedure and sample size.

We are grateful for the Reviewer’s constructive comments on our manuscript. We have now addressed the issues raised by the Reviewer, and provide detailed responses, below.

Line 140-145 indicates that the authors used saline swabs for (the first?) 13 days of project, but then switched to vaginal lavage, since the swabs yielded “cell densities that were sometimes too low for oestrus stage identification” (l. 141-141). Later in the results, the authors indicate that saline swabs were used for days 1-10 (not for 13 days as indicated in l.144), and lavage for days 11-27. However, they also indicate that data from swabs were discarded (l. 355-356). Were all discarded? If so, why is

the sampling period still represented as 27 days (l. 136)? If no data in the manuscript were from these samples, then why is this method included at all? The authors should clarify whether data presented in the manuscript were gathered from two different methods, and if so, provide evidence that they produce equivalent results. Could the lavage yield different types of cells than the swab, for instance by sampling the vaginal cavity more widely or deeply? Is the risk of cervical stimulation and pseudopregnancy (re: l. 139) different (l. 452 notes a problem of prolonged oestrus stages in the finding but doesn't elaborate)?

We apologize for the lack of clarity in explaining our sample collection process. While our mice were kept for 27 consecutive days, they were sampled for 17 consecutive days for vaginal cytology assessment. We collected urine on all 17 days, but we only conducted odour analyses for a subset of these days (i.e. between day 1 and day 17, inconsecutively). The reason for this was due to limitations in time and funding (a single researcher collected all the samples and carried out the cytology study), and technical difficulties following the malfunction of our first Torion instrument. We have now revised our manuscript to only mention the 17 consecutive days for which cytology assessment was carried out, in order to clarify our sampling.

We have also updated the supplementary table S1 and our Methods section to clarify the range of samples collected for each mouse, both for vaginal cytology assessment and for urinary VOC analyses: “we collected one vaginal sample from each mouse for 17 days at approximately 13:00 (suppl. table S1). Vaginal smears were collected onto microscope slides using saline swabs [36] (days 1 – 2) or saline lavage [37] (days 3 – 17). With the swab method, cotton swabs dipped in saline solution were inserted 1-2 mm into the vaginal canal. During the course of our experiment, we found that the vaginal smears prepared from saline swabs were suboptimal in terms of cell density relative to saline lavage. We used saline swabs for two consecutive days before we exclusively used saline lavage to prepare the vaginal smears.” (lines 144-150);

“We selected a representative subset of samples for VOC analysis using portable GC-MS. We analysed a total of 90 urine samples (including all 40 of the samples that were also analysed on the benchtop unit) using headspace extraction on the PerkinElmer Torion® T-9 portable GC-MS instrument [43] at the University of Calgary, AB, Canada (suppl. table S1).” (lines 254-258);

“Due to constraints on time and funding for analyses, we analysed a subset (40) of the urine samples collected from each mouse, using an Agilent benchtop GC-MS system (suppl. table S1).” (lines 225-227).

We have also clarified our Results: “We collected vaginal smears for 17 consecutive days. Throughout day 1 – 2, saline swabs were used to prepare vaginal smears. From day 3, we switched to saline lavage instead of saline swabs to prepare vaginal smears as the vaginal smears prepared from saline lavage yielded greater consistency in smear quality.” (lines 393-396).

We apologize for not commenting on the risk of inducing pseudopregnancy resulting from cervical stimulation when employing saline lavage and swabbing for cytology assessment. We employed careful handling and conservative insertion of special small-sized tapered

swabs and pipette tips into the vaginal orifice to avoid cervical stimulation. We did not find any evidence of pseudopregnancy, which appears as prolonged dioestrus stage for up to 14 days. Rather our cytological findings that were documented for 17 consecutive days indicate regular cycling parameters.

We have added details on this in the Methods: “Care was taken to avoid cervical stimulation during both methods (saline swabs and saline lavage), which can lead to pseudopregnancy, which appears as prolonged dioestrus for up to 14 days [1,14]. The induction of pseudopregnancy by cervical stimulation is unlikely when samples are collected with care [14]. Our results indicate regular cycling for the entire duration of the study, and we see no evidence of prolonged dioestrus (suppl. table S1).” (lines 153-157).

To compound this problem further, the supplementary material also shows different sampling days from the main text. Table S1 indicates that urine samples from days 1-17 were analyzed, contrasting with either the number and/or labeling of days from the two different timelines in the main text. Lastly, Table S2 appears to count fully through days 1-29 (not 27) in the cells, even though the header suggests only 17 consecutive days were sampled. Overall, it is ambiguous for readers how many days mice were sampled for, and whether or not data from these sample days were collected with two different sampling procedures.

We agree with the Reviewer’s point, and we are sorry for this confusion. We have replaced Tables S1, S2 and S3 with an updated Table S1 which clearly states the days of collection of vaginal smears and urine, and those of urinary odour analysis, for each mouse.

We have clarified the difference between the number of days for which vaginal smears and urine were sampled, and those for which odour analyses were carried out, in the main text. Please also see our response to the comment above.

Methods: “Due to constraints on time and funding for analyses, we analysed a subset (40) of the urine samples collected from each mouse, using an Agilent benchtop GC-MS system (suppl. table S1).” (lines 225-227);

“We selected a representative subset of samples for VOC analysis using portable GC-MS. We analysed a total of 90 urine samples (including all 40 of the samples that were also analysed on the benchtop unit) using headspace extraction on the PerkinElmer Torion® T-9 portable GC-MS instrument [43] at the University of Calgary, AB, Canada (suppl. table S1).” (lines 254-258).

Results: we have removed mention of urine in section 3.1. (line 393), as this section was about the cytology results;

We have also removed the ambiguous sentence: “We discarded the data prepared from saline swabs (day 1-10) and assessed the vaginal smears prepared from saline lavage (day 11 - 27) for each mouse.” (line 394).

Discussion: We have replaced “samples” with “urinary odour samples” (line 609).

There are also inconsistencies in reporting the analyzed sample size, beyond the number of days sampled. L. 190 indicates that urine collection occurred following vaginal cytology sampling, implying an equal number of urine samples (for odorants) and cytology samples (for oestrus classification). If either days 1-10 or 1-13 of the swab technique were discarded (re: l. 144 or l. 356) from the 27 total days, then this should be N=170 (17 days x 10 mice) or N=140 (14 days) samples. However, l. 206 & 232 indicates that only N=90 urine samples were analyzed by GC-MS (N=40 by both the benchtop and portable unit, and an additional N=50 by the portable unit). Likewise Table S1 suggests that only certain days were sampled for each mice, as does l. 557 of the discussion. As far as I can ascertain, there are no details provided on whether this reduced sample size was due to inconsistent gathering of samples, or whether only a certain subset of samples were chosen for analysis, and why.

We apologize and agree this was confusing. We have now made changes to the main text and updated our suppl. table S1 to clarify information on the range of samples collected for each mouse, both for vaginal cytology assessment, and for urinary VOC analyses. Please also see our responses to comments, above.

This manuscript would also benefit from better situating the context and novelty of its aims within the available literature. Both of the aims of the study, to 1) characterize mouse odorants across estrus cycles, and 2) compare the efficacy of portable and benchtop GC-MS units, have been studied before, in literature cited by the authors. However, the introduction does not provide this important piece of background for readers. Notably, the Andreolini *et al.* (1987) citation (l. 653) contains data very similar to current study, having both the same aims and methods, in which vaginal cytology is used to analyze mouse oestrous stages in relation to odorant compounds from urine analyzed with GC-MS. Given this, what are the real novel aims of this manuscript beyond this previous work? Is it technological advancement in GC-MS since 1987 that warrants reevaluation of these data? Or was this previous research somehow deficient and/or in need of replication? GC-MS has certainly advanced in the 20+ years since Andreolini *et al.* (1987), and the current manuscript does report much more detailed findings. Nonetheless, it is important for readers to be aware that this research aim has been previously conducted, and the results of each should be directly compared, particularly since the Andreolini *et al.* publication DID find differences in odorants between oestrus cycles, while the current manuscript did not. Overall, the introduction leaves readers with the impression that we know little about odorants in mice or how they vary with oestrus. However, the literature presented in the discussion (l. 564-571) suggest that this is not entirely the case.

We thank the Reviewer for this suggestion. We feel that the strengths of our study include the more detailed findings, also highlighted by the reviewer, and the direct comparison with a new, portable GC-MS. We agree, however, that we should do a better job of situating our research with existing studies. Accordingly, we now draw attention to Andreolini *et al.* in our Introduction: “A number of volatile and semi-volatile chemicals indicators of reproductive status in mouse urine have been identified [19–21]. Andreolini and colleagues [19] reported 28 compounds in the urine of female mice using GC-MS. Of these compounds, 11 volatiles varied in concentration across different stages of the oestrous cycle.” (lines 72-76).

We have added mention of previous research on the topic in our aims & hypotheses: “The chemical classification of these compounds will provide new fundamental knowledge

regarding the substrates of communication and possibly sexual selection, and provide valuable comparison with existing research on murine urinary scents.” (lines 115-118).

This is also true for the second aim, comparing portable to benchtop GC-MS devices, albeit to a lesser degree. This comparison has been previously been performed by Kücklich *et al.* (2017) and Poirier (2019). The current authors do compare their results to these previous studies in the discussion, but do not mention that this comparison (with different device models) has already been done in the introduction, when stating their goals. Again, what is the exact utility here—to compare a different model of portable GC-MS? A different run time? Extraction method? Being more exact about how the current study builds on previous findings would better equip the reader to utilize the information in this manuscript. Overall, the introduction leaves readers with the impression that these aims have not previously been studied, which is not supported by the literature.

We agree with the Reviewer, and have now added further details about the novelty and goals of the current study in the Introduction. These details build on our previous description of the two existing studies on mammalian scent analysis using portable systems by Kücklich *et al.* and Poirier *et al.*: “In particular, portable devices have not yet been employed for the analysis of urinary VOCs; nor have they been used for repeated analysis of odours over time, such as across the female reproductive cycle, or directly compared to benchtop models using the same samples. These comparisons will help assess the utility of different equipment” (lines 99-103).

Our Discussion section 4.2 also provides further arguments for the novelty of the present study: “To the best of our knowledge, this was the first use of the chromatoprobe VOC traps for the extraction of urinary VOCs.” (lines 551-552); “Our study also presented the first use of the Torion® portable GC-MS for the analysis of urinary VOCs.” (568-569).

Line by line comments

l. 30-31 (+ l. 481-483) This direct comparison implies to readers that the samples are expected to be identical when analyzed on different devices...but is this assumption really true? Nair *et al.* (2018) has a nice discussion of the idea that of the many different methods available to sample odorants, each is impacted by collection methods, media for preservation (with differing retention biases), and analysis devices (that differ in the detection abilities) that will impact the type of compounds that are retained and detected. In short, while all methods are sampling the same population of compounds, the samples derived from each will inherently differ, based on different collection and storage procedures, and the inherent capacities of the GC-MS devices there are analyzed on. Comparing the samples obtained from two devices and methods used in the manuscript is a worthwhile endeavor. However, several places in the text seem to present this as a direct ‘apples to apples’ comparison—with the implicit assumption that samples obtained should be the same, despite that this does not really portray the nuanced situations of these differing methods.

Nair, J. V., Shanmugam, P. V., Karpe, S. D., Ramakrishnan, U., & Olsson, S. (2018). An optimized protocol for large-scale in situ sampling and analysis of volatile organic compounds. *Ecology and Evolution*, 8(11), 5924–5936.

We agree with the Reviewer about the limitations in directly comparing the GC-MS systems used in this study, when VOC sampling was also carried out using two different techniques. Nevertheless, even different sampling and analytical techniques could have led to the detection of the same types of compounds, albeit in different abundance. The fact that only three compounds were common to the two techniques was a surprising result, worth mentioning in our opinion. Moreover, we believe it made sense to compare the overall number of compounds detected with each system. For this reason we felt it was valuable to highlight this in the abstract (lines 35-37).

Our Discussion section 4.2 now expands on the differences in sampling capacities between SPME and VOC traps, and on the differences in compound separation capacities between portable and benchtop systems: we have added a summary statement to improve the clarity of our argument: “The difference observed between benchtop and portable systems in the aforementioned studies, as well as in our study, was expected to some degree, since the extremely short GC run in a portable instrument” (lines 537-541);

We have also added a sentence at the end of the paragraph to better acknowledge the difficulty of drawing direct comparisons between the two systems used here: “The differences in VOC sampling, extraction and detection capacities between the two systems used here made drawing direct comparisons difficult [93].” (lines 547-549).

I. 32 It would help readers to know why these particular compounds are ‘notable’ here.

We have added a sentence to further explain our point: “These VOCs may be indicators of mouse fertility” (lines 33).

I. 37 What exactly is meant by ‘subtle’ here? Low abundance of certain compounds? A limited number of compounds that vary between stages? A more exact description would help readers.

We have replaced the sentence with “It is possible that the changes in VOC abundance were too small to be detected by our analytical methods.” in the Abstract (lines 39-40).

I. 73-75 As indicated above, I think it would help readers to more fully describe what is known about mouse urine odorants (re: the literature cited in l. 565-569).

We agree and have updated our introduction to include more background knowledge about the chemical composition of mouse urine and variations with the oestrous cycle: “A number of volatile and semi-volatile chemicals indicators of reproductive status in mouse urine have been identified [19–21]. Andreolini and colleagues [19] reported 28 compounds in the urine of female mice using GC-MS. Of these compounds, 11 volatiles varied in concentration across different stages of the oestrous cycle.” (lines 72-76).

I. 90 Update “in review” or cite thesis, as appropriate.

We have updated the Poirier *et al.* citation (line 94).

Poirier, A. C., Waterhouse, J. S., Watsa, M., Erkenwick, G. A., Moreira, L. A. A., Tang, J., ... Smith, A. C. (2021). On the trail of primate scent signals: A field analysis of callitrichid scent-gland secretions by portable gas chromatography-mass spectrometry. *American Journal of Primatology*, 83(3), 1–12. <https://doi.org/10.1002/ajp.23236>

I. 104-105 “These data” and also, this should be revised with the above comments. Given that research on this topic has already been done, what further advances in husbandry does the current study provide?

We thank the Reviewer for raising this. We have corrected the typo (line 110).

We agree with the Reviewer that our study does not provide direct advances in mouse husbandry, so we have removed the end of the sentence (line 111). Rather, our study provides valuable comparison with existing research on murine urinary scents (lines 115-118).

I. 112 Recommend delete “proof-of-concept” or greatly refine what is meant here. Kücklich *et al.* (2017) conducted a similar, direct comparison of portable and benchtop GC-MS, seemingly already “proving the concept” demonstrated in the current manuscript. It is unclear that the captive methods here, particularly daily urine samples and vaginal swabs/lavage, would be possible for most wild field studies.

We agree with the Reviewer; we have replaced the term “proof-of-concept” with “new” (line 119).

I. 135-137 As indicated above, it is unclear whether mice were sampled for 27 days, and/or whether 27 days of data are represented in the sample.

We agree this was previously confusing. We have cytology data for 17 consecutive days, and odour data for a subset of these days, as detailed in suppl. Table S1. We have revised our text and supplementary table to remove the confusion. Please also see our responses to previous comments.

I. 140-145 As indicated above, it needs to be clarified whether the data from the swabs are present in the current manuscript. If so, then evidence that these methods produce equivalent results, and can be appropriately pooled, should be provided.

We agree that the inclusion of data generated from swabs needs to be clarified. We have included the data from swabs collected on days 1 and 2 (suppl. table 1). Oestrous stage identification of the smears that originated from swabs collected on days 1 and 2 were

categorized by discerning cell composition and morphology. Stages were further scrutinized based on the succeeding days' stages for accurate identification of the cycling status. Data from days 1 and 2 are consistent with the following days' data, which were generated from saline lavage (suppl. table 1). As detailed in a prior comment from Reviewer 2, we have updated suppl. table S1 and made improvements throughout the manuscript in order to enhance clarity about our sample size.

I. 161+ I recommend providing a citation for the methods to classify oestrus status.

We thank the Reviewer for the suggestion. We have now cited Cora et al. 2015 (line 181).

I. 169-170 Is this a standardized method, or a new one? There has been much literature on categorizing mouse oestrus, it would be helpful to know if the authors are following established recommendations.

We have clarified that we followed the methods established in Cora et al 2015, cited line 181.

I. 182-186 Please add acronym definitions for the transitional stages too. If possible, I think a color figure would be more readily interpreted by readers.

We have added definitions for the transitional stages: “The transitional stages pro-oestrus–oestrus (PE), oestrus–metoestrus (EM), metoestrus–dioestrus (MD) and dioestrus–pro-oestrus (DP) show characteristics of both prior and later stages.” (lines 198-200).

We greatly appreciate this suggestion and agree that a colour figure would enable readability. Unfortunately, the light microscope (Zeiss Axio Vert.A1) only captured grayscale images of the monochromatically stained slides. We hope that the different cell types and morphologies remain discernable for readers despite the absence of colour.

I. 193-194 Please provide the mean, SD and range for both the days until analysis and drops of urine. Additionally, evidence should be provided on the stability of VOC held for this long. Even when frozen, some compounds have been shown to degrade within months.

We have added the requested information on the number of days until analysis and the number of drops collected, and addressed the problem of storing urine for long periods: “The vials were stored at -20°C for an average of 129.2 ± 37.3 days (mean \pm SD; range 100 – 205) until analysis. Keeping urine frozen limited the degradation of VOCs owing to bacterial activity, although it is possible that some VOCs were lost during this prolonged storage [40]. The volume of urine contained in each vial varied between mice (9.6 ± 8.0 drops; range 2 – 40).” (lines 208-212).

I. 201-202 It would be helpful to add a regression of the relative peak area and sample volume to the supplemental material. This is also used as a variable in the authors' analyses, and there is some reason to believe that this measure of abundance may be more impacted by volume than simple presence/absence of compounds.

We are grateful for the Reviewer's suggestion. We have updated our supplementary figure S1 to include results of the regression of the relative peak area and sample volume; these were also non-significant for both instruments.

We have updated the sentence referring to suppl. figure S1 in the main text: "Because the range of urine volume sampled here was quite important (i.e. a 20-fold variation), we further verified a lack of relationship between sample volume and our variables in the analyses using linear regressions (suppl. figure S1)." (lines 218-220).

I. 197-200 Please add a citation for samples being robust to volume changes, as well as some definition of what is meant by "small difference." The urine samples in this study are all low volume, but the difference between the smallest volume, 2 drops, and the largest, 40 drops, is a 20-fold increase.

We agree with the Reviewer that the difference in volume of urine sampled was not small. We have referenced the following statement: "Our sampling procedure is robust against small differences in sample volume, when the samples are allowed to equilibrate and are measured at the same standardised temperature (i.e. 21°C), since in standardized conditions vapor pressure will vary only with sample concentration, not volume." (lines 215-218).

I. 211 Please provide a range, mean, or estimate for "small volume." Was this standardized?

We have added details to the manuscript: "(i.e. two drops sampled with a glass Pasteur pipette, ca. 0.44 mL)" (line 232).

I. 215-218 15+ months at 8°C is quite some time for an analysis of highly volatile odorant compounds. Studies on human urine have indicated that even samples frozen at -80°C should be analyzed within 9 months to prevent degradation (e.g. Esfahani et al., 2016). Evidence should be provided to support the claim that degradation in chromatoprobes does not occur over this timeframe.

Esfahani, S., Sagar, N. M., Kyrou, I., Mozdiak, E., O'Connell, N., Nwokolo, C., ... & Covington, J. A. (2016). Variation in gas and volatile compound emissions from human urine as it ages, measured by an electronic nose. *Biosensors*, 6(1), 4.

The temperature given was erroneous, we apologize for this. The accurate temperature of storage was -20°C (line 238).

We have added information on stability of the chromatoprobe VOC traps in the Discussion: "Chromatoprobes show extremely high shelf stability, with the signal retaining its strength even after >3 years and with transport from remote tropical stations and no cooling (Nevo,

unpublished data). We are therefore confident that given the short storage time and continuous cooling, no significant amount of signal was lost.” (lines 563-566).

We have also added mention of the VOC trap system in our Conclusion, as we feel this was a promising method: “Chromatoprobe VOC traps appear well suited for the sampling of mammalian urinary volatiles. They constitute a light-weighted, versatile and stable sampling method, readily applicable to field studies.” (lines 656-658).

I. 232-233 For the samples measured with both methods, which analysis was done first, and was this standardized? L.195-196 indicates that opening the sample tube generates “loss” of VOCs. Systematically conducting one analysis first or second could create bias due to lost VOCs from the previous opening.

We always carried out the headspace analyses using the SPME fiber first, by puncturing the membranous septum covering the glass vial. Thereby, we did not open the vial until we needed to transfer urine into the Toppits bag for chromatoprobe analysis.

We have added this information to the Methods: “We selected a representative subset of samples for VOC analysis using portable GC-MS. We analysed a total of 90 urine samples (including all 40 of the samples that were also analysed on the benchtop unit)” (lines 254-256);

“We then exposed a 65 µm PerkinElmer Custodion® polydimethylsiloxane/ divinylbenzene (DVB/PDMS) SPME fibre [43] to the headspace of each sample for 2 min, through the vial septum (vials were not opened, thereby avoiding the loss of VOCs).” (lines 259-262).

We have also made clear that only a subset of samples were analysed using the benchtop system: “Due to constraints on time and funding for analyses, we analysed a subset (40) of the urine samples collected from each mouse, using an Agilent benchtop GC-MS system (suppl. table S1).” (lines 225-227).

I. 241 For Supp Mat, Table S1, it would be helpful to provide data from the benchtop in the table, to better compare samples analyzed both ways.

We thank the Reviewer for this suggestion; we have now added data from the benchtop analyses to suppl. table S1.

I. 245-249 Were blanks and controls also run on the benchtop system? How were compounds found in controls treated in the results?

We did collect control samples with the chromatoprobe technique. We have added it to the Methods: “Control samples (i.e. empty bags) were additionally collected in the same conditions.” (lines 236-237).

We treated results from the benchtop and the portable instruments in the same way; we have updated our Methods to improve clarity: “We removed compounds whose identity was clearly inorganic, e.g. silane derivatives, from the remaining peaks, we only selected those with a minimum area of 0.01% of the chromatogram’s total signal. This step allowed us to limit the inclusion of background noise, as peaks under this threshold were generally too flat to be distinguished either from the baseline or from a neighbouring peak. In addition, we removed from further analysis all peaks found in higher amounts in at least one blank sample.” (lines 299-305).

I. 298 It would be helpful to provide a brief explanation of ‘oestrus cycle’ and why this variable is important in models.

Our updated suppl table S1 clearly shows the succession of oestrous cycles throughout the duration of the study. For repeated sampling, as was the case here, it is common to include a temporal variable as a random effect in linear models. This can be the day, year or season of collection, or in our case, the oestrous cycle ID.

For improved clarity, we now refer to this variable as “oestrous cycle ID” instead of “oestrous cycle” throughout the text.

I. 352-357 As indicated above, if all swab data were discarded, then this should probably be deleted. In the least, how these data were treated should be addressed earlier in the methods.

The saline swab vaginal smear technique was used on the two first days of urine collection for all mice, therefore we could not justify omitting the mention of this technique. As explained above, we have revised our manuscript and supplementary material to better clarify this ambiguity.

Table 1 & 2. Based on the references provided, these tables appear to omit data on humans, despite that this is one of the richest sources of data on mammalian urinary VOCs. I also recommend referencing Charpentier et al. 2012, whose supplemental material has a comprehensive review of compounds used in mammalian communication.

Charpentier, M. J. E., Barthes, N., Proffit, M., Bessière, J.-M., & Grison, C. (2012). Critical thinking in the chemical ecology of mammalian communication: Roadmap for future studies. *Functional Ecology*, 26(4), 769–774. <https://doi.org/10.1111/j.1365-2435.2012.01998.x>

We thank the Reviewer for raising this deficiency. We have now added references from Mochalski et al. 2012, Smith et al. 2008, Wahl et al. 1999, and Wagenstaller & Buetter 2013 on humans, as well as a recent reference on non-human primates (Caspers et al. 2020), in Tables 1 and 2.

We have also referenced Charpentier et al. 2012, an excellent review, in our Discussion (line 626).

I. 421 Please state the “metric.” Is this the Bray-Curtis index?

We have added this information to the sentence: “Similarly, our metric of the chemical composition of samples, the Bray-Curtis dissimilarity index, did not significantly differ between oestrous stages (table 3).” (lines 462-463).

I. 454-455 Light cycle was controlled in this study (l. 126), so it doesn’t seem like a very pertinent explanatory factor for these findings.

We agree with the Reviewer and have removed this sentence (line 502).

I. 465 Missing stages do not appear to be reported in the results. While mean and SD of stage length is given, there is not a mention of overly prolonged stages in the results, either.

We have added mention of missing and prolonged stages in the Results: “Though we collected smears at approximately the same time every day, missing stages may be explained by stage length variation.” (lines 396-397).

We have also discussed it further in the Discussion: “The occasional occurrence of prolonged oestrus stage and absence of other stages resulted in variable (3 – 6 day) cycle lengths. Cycle length variability and the absence of some stages may be explained by natural variation between individual mice and our sampling frequency, respectively. Oestrous stages can range between 6 – 72 hours depending on the stage and mouse [1]. We collected vaginal smears once every 24 h, which could have resulted in missing stages that were shorter than 24 h.” (lines 497-502).

I. 468 Reference #73 is from 1941, and references 25 & 26 are from 2012 and 2013, respectively. Given this age, why were these optimal methods not already incorporated in this study? Presenting these as a way to improve data collection in the future does not seem appropriate.

We thank the Reviewer for this comment. Limitations in time and funding did not allow us to employ these methods in our study. We chose to present these methods for investigators who wish to further optimize the quality of smears through the use of multichromatic stains. The hue of these cells vary with the degree of cornification and may be useful to those who wish to fully appreciate the nuances of morphological and compositional changes that occur across the rodent estrous cycle. As crystal violet is often used for bacterial staining, it is more affordable when compared to Schorr and Papanicolaou stain. However, multichromatic stains such as Papanicolaou stain, remain great options for those who wish to assess their smears in greater detail, which is why we have mentioned them in our manuscript (lines 515-517).

I. 481-483 This should be expanded on. These instruments have different sampling capacities, and sampling and extraction procedures differed between the two protocols. Since these procedures can

impact which compounds are retained, and the devices themselves have different capacities to detect compounds, should these results really be the same? I have no problem with the idea of comparing the results obtained from portable and benchtop models, but I think this text could better portray for readers that there are many reasons that these sampling methods are not able or expected to produce similar results.

We thank the reviewer for this important comment. Our Discussion section 4.2 now expands on the differences in sampling capacities between SPME and VOC traps, and on the differences in compound separation capacities between portable and benchtop systems.

We have added a summary statement to improve the clarity of our argument: “The difference observed between benchtop and portable systems in the aforementioned studies, as well as in our study, was expected to some degree since the extremely short GC run in a portable instrument (i.e. around 3 min in our study, in comparison to the 56 min runtime with the benchtop instrument) does not allow complete separation of individual chemicals over time.” (lines 537-541).

We have also added a sentence at the end of the paragraph to better acknowledge the difficulty of drawing direct comparisons between the two systems used here: “The differences in VOC sampling, extraction and detection capacities between the two systems used here made drawing direct comparisons difficult [93].” (lines 547-549).

I. 504-506 Further information should be provided on the shelf stability of compounds gathered in chromatoprobes.

We have added information on stability of the chromatoprobe VOC traps: “Chromatoprobes show extremely high shelf stability, with the signal retaining its strength even after >3 years and with transport from remote tropical stations and no cooling (Nevo, unpublished data). We are therefore confident that given the short storage time and continuous cooling, no significant amount of signal was lost.” (lines 563-566).

I. 544-545 Does this capacity differ between the two devices?

We have expanded the argument to report results from each device: “we identified VOCs with a molecular weight between 58.1 (propan-2-one) and 450.9 g/mol (dotriacontane) using VOC trap thermal desorption on the benchtop device; this range was lower when using SPME on the portable device.” (lines 595-597).

I. 557 I believe this is the first mention that samples were not collected every day, and adds to the confusion of the sampling procedure and sample size. There need to be explicit methods given on which days mice were samples, which sample were analyzed and why.

We agree with the Reviewer, and our updated suppl. table S1 now provides improved clarity on which days were sampled for urinary VOCs across the 17 days of cytology assessment. Please also see our previous answers on the same topic, above.

I. 564-571 This body of literature, particularly Andreolini *et al.* (1987), should be more thoroughly covered in the introduction.

We agree and have acknowledged the work by Andreolini *et al.* more extensively in our Introduction (lines 74-76).

Supplementary Table S1. Adding a data column for the number of VOCs from the benchtop analyzed samples would be helpful. Why were some days not sampled for mice, and why does this differ between mice? Were control and blank compounds removed from these tallies?

We have added the number of VOCs detected using the benchtop system to suppl. table S1. As detailed above, we only conducted odour analyses for a subset of the 17 days for which vaginal cytology was assessed.

We have revised the caption of table S1 in order to improve clarity about the samples collected and the number of compounds detected: “Vaginal samples used for the determination of female mice oestrous stage across 17 days on ten animals (designated by alternating cell color shading); urine samples used for the detection of VOCs at the different stages of oestrous; and number of VOCs detected in urinary samples. Urinary samples were analysed for a subset of days via two GC-MS methods: 1. extraction by chromatoprobe thermal desorption and analysis on an Agilent benchtop GC-MS; and 2. extraction by SPME and analysis on a Torion® portable GC-MS, using two different instruments (i.e. Torion 1 and Torion 2) and two different runtimes (i.e. 190 s and 170 s). Sample choices were impacted by labour and funding limitations, and constrained by the malfunction of one of the portable instruments.”

Supplementary Table S2. The ‘days’ headers are confusing. Why are there two overlapping headers for day, and why are days 13-17 both on the first and second line? Why are there 29 days in column headers, but 17 indicated in the caption?

We have removed this table from our supplementary material. The information it contained is now visible in suppl. table S1, which is more comprehensive.

Appendix B

Response to Reviewers – Tang et al. RSOS – Resubmission July 2021

Response to the Editor's and Reviewers' comments

Subject Editor Comments to Author (Professor Kevin Padian):

Comments to the Author:

Thanks for your revision. The concerns that the second reviewer raises about the preservation of the sample, which appear to differ from usual standards, and how that may have affected your results, are serious ones that in the reviewer's judgment have not been adequately addressed previously. Please re-assess and rewrite to take these into consideration, and address them in a separate document upon resubmission. We regret that we will not be able to send your MS out for another round of review, but this reviewer may be asked to look at it again. Best wishes for your revisions and if you need more time please let the editorial office know.

Dear Professor Padian,

We are grateful for the opportunity to address the Reviewers' constructive comments, which we have done in detail below. Our manuscript has been improved in several ways. In our revised manuscript we also better acknowledge the long storage time of the urine samples prior to odor analyses and that this is a limitation of our study. We also provide a more extensive explanation, including references, for why we believe the chromatoprobe VOC traps are robust even given long term storage. We additionally better acknowledge this as a limitation and also better explain why we believe the probe storage did not significantly affect the interpretation of our results. We hope our manuscript is now found suitable for publication at Royal Society Open Science.

Best regards,

Jia Tang and Alice Poirier, on behalf of all co-authors

Reviewer comments to Author:

Reviewer: 1

Comments to the Author(s)

The manuscript has improved, but few minor issues still have to be resolved:

1) Page 8, Line 43: do authors mean CIS4 instead of CIC 4C?

We thank the Reviewer for the supportive comment and additional suggestions and feedback. Our apologies for this typo - the Reviewer is correct and we have made the suggested change (line 285).

2) Page 8, Line 45: “Samples were introduced splitless to the thermal desorption unit (TDU) at 30°C” – maybe samples were desorbed in the thermal desorption unit in splitless mode? What was the flow rate in TDU during desorption?

We thank the Reviewer for the suggestion that improves the clarity of our manuscript. We have adopted this suggestion, as shown below. In addition, we now also provide a full GC-MS methods file as an example in our supplementary material, to ensure full transparency and reproducibility (lines 286-288):

“Samples were introduced to the thermal desorption unit (TDU) at 30°C in splitless mode, using helium as the carrier gas at a flow rate of 1 mL/min. A full GC-MS methods file is provided as an example analysis protocol in suppl. table S2.”.

To avoid repetition, we have now deleted the following mention of carrier gas later on in the paragraph, line 293: (“We used helium at a flow of 1 mL/min as carrier gas”).

3) Page 8, Lines 53-55: “We used helium at a flow of 1 mL/min as carrier gas.” - what was the flow rate of He in the liner during desorption of analytes from liner to the column? Was the CIS in splitless mode during this step?

We have now provided this information in the form of a methods file from our benchtop GC-MS analyses, which we reference as supplementary table S2 (lines 286-288):

“Samples were introduced to the thermal desorption unit (TDU) at 30°C in splitless mode, using helium as the carrier gas at a flow rate of 1 mL/min. A full GC-MS methods file is provided as an example analysis protocol in suppl. table S2.”

4) Page 9, Line 16: better used commercial fiber title: PDMS/DVB instead of DVB/PDMS

We thank the Reviewer for this advice. We have made the requested change from DVB/PDMS to PDMS/DVB (lines 236 & 238).

Reviewer: 2

Comments to the Author(s)

I appreciate the opportunity to once again review manuscript RSOS-210172. The authors have made a number of revisions that have improved the quality of the manuscript. This includes refining the background information given to readers, clarifying the novelty of the study, and reporting the sample size and sampling days in a more straightforward manner. However, there are still a few remaining items that I do not believe have been satisfactorily addressed.

We thank the Reviewer for their constructive feedback. We apologize that we did not provide adequate information in the first revision and we now address each of the points in more detail, below.

1. First, the data collection methods for vaginal cytology were switched part way through the study, but the manuscript does not provide any evidence that the two methods (swab and lavage) produce equivalent results. While this version better clarifies for readers what method was used when, there is still no assurance that the methods produced consistent results. I recommend the authors either: 1) cite references to support the equivalency of these methods, 2) provide any possible supplemental data from the current study that demonstrates equivalency, or 3) delete the two days of data collected by swabs, so that all data are from the lavage method.

We thank the Reviewer for providing these suggestions. We now more directly support our assertion that cytological assessment from swab and lavage methods give similar results in two ways:

1) We have clarified our text and citations supporting the equivalency of these two methods, as below. Importantly, we explain that with both methods we were able to clearly identify the cell types that allow us to assign cycle phase with confidence,

Lines 143-153: “The division of the oestrous cycle into the four main stages of pro-oestrus, oestrus, metoestrus, and dioestrus, is defined by the presence, proportion, density, and arrangements of four basic cell types. Collection of the samples can be reliably done either by vaginal lavage or swabbing [1] and both methods have been commonly used [14,15,37]. We found that both methods yielded high-quality sample slides that contained all of the characteristic cells of the vaginal epithelium necessary to assign oestrous stage (suppl. figure S1). We initially started with the swab method, as it was slightly less technically challenging, but we switched to the lavage method as it was less invasive and we found, consistent with Cora et al. [1], that lavage yielded a higher cellularity sample than samples collected by swabbing the vagina, thus making assignment of stages easier.

Lines 158-160: “With the swab method, cotton swabs dipped in saline solution were inserted 1-2 mm into the vaginal canal. With the lavage method, approximately 0.1 mL of saline solution drawn into a pipette was inserted into the vaginal orifice at a depth of 1-2 mm.”

We have also correspondingly revised section 3.1. of the Results to remove information about the equivalence of the cytology methods used, since it is now well covered in the Methods. We have removed: “Throughout day 1 – 2, saline swabs were used to prepare vaginal smears. From day 3, we switched to saline lavage instead of saline swabs to prepare vaginal smears as the vaginal smears prepared from saline lavage yielded greater consistency in smear quality.

Though we collected smears at approximately the same time every day, missing stages may be explained by stage length variation.”

The revised sentence is as follows, lines 402-403: “Examining vaginal smears collected from 10 mice over 17 consecutive days revealed that the mice were not cycling in unison with each other.”

2) Furthermore, as Cora et al (2015) describe, accurate interpretation of vaginal cytology samples is dependent on the quality of the sample preparation, regardless of the methodology (swab or lavage) employed. To demonstrate that both our methods yielded high-quality sample slides, we have now included additional images in the supplementary material as suppl. figure S1. This figure compares smears prepared by the two methods from the same oestrous stage collected on different days, to demonstrate that we achieved the quality necessary to see the cell types and accurately categorize the oestrous stage using each method. We did not conduct two cytology assessment methods on the same day for the same mouse as this might have caused excess stress and was not in our ethics protocol. However, when we compare the data collected by the two methods at the same stage, we see smears of comparable cell composition, morphology, and density: pro-oestrus smears collected both comprised of characteristic nucleated epithelial cells; oestrus smears collected contained cornified epithelial cells; metoestrus smears collected consisted of neutrophils and nucleated and cornified epithelial cells; and dioestrus smears had similar cell composition to metoestrus smears but in differing proportions (suppl. figure S1).

Oestrous stage data prepared by swabs (i.e. from the first two days of collection) were further supported by the classification of succeeding stages from the slides prepared by lavage. For instance, the classification of oestrus for mouse M01 using the swab method on day 1 and 2 was supported by the succeeding metestrus classification on day 3 using the lavage method (suppl. table S1).

Suppl. Figure S1. Vaginal smears viewed at 100 x magnification, showing pro-oestrus prepared using a) the swab method and b) the lavage method, for mouse M10; showing oestrus prepared using c) the swab method and d) the lavage method, for mouse M01; showing metoestrus prepared using e) the swab method and f) the lavage method, for mouse M03 and; showing dioestrus prepared using g) the swab method and h) the lavage method, for mouse M09. The two cytology methods allowed the correct identification of the different oestrous stages.

2. Second, the samples in this study were stored for much longer, and at a warmer temperature, than recommended by references cited by the authors. Yet, several points in the manuscript's text suggest that this storage period is not problematic, or downplay the time frame (pg. 7 l. 45-50, pg. 8 l. 35-37, pg. 17 l. 46-51, pg. 20 l. 19-23). This assertion is inconsistent with the literature. Saude & Sykes (2007), which is cited by the manuscript, shows severe degradation of VOCs after four weeks when stored at 4°C, and consistent (although much less severe) degradation after 4 weeks when stored at -80°C. By contrast, the current manuscript

stored urine at -20°C for 14-29 weeks (100-205 days; pg. 7 l. 46—and possibly an additional year in chromatoprobes; see below comments). Likewise, and as pointed out in my previous review, Esfahani et al. (2016) recommended that samples be frozen at -80°C and analyzed within 9 months. Overall, the storage of odour samples does not appear to be in line with current guidelines. Based on the previous literature, it seems likely this study experienced degradation of samples. This needs to be better acknowledged for readers, the text should be modified to more accurately portray the literature on VOC storage degradation, and appropriate context for the storage times frames used in this study, relative to the published literature, needs to be provided.

We thank the Reviewer for this comment; we agree that our manuscript did not fully acknowledge the potential limitations of our work owing to the prolonged storage of urine, which we have now corrected.

We have revised our manuscript accordingly. We have removed unjustified statements about whether storage was problematic from the Methods (removed: “Keeping urine frozen limited the degradation of VOCs owing to bacterial activity, although it is possible that some VOCs were lost during this prolonged storage [40].”, line 210). In our revised sentences we better explain and state our storage conditions and we clarified a key point that all samples were treated similarly.

The revised Methods is as follow, lines 205-209:

“Due to delays caused by the portable GC-MS needing to undergo servicing, we stored the vials prior to analysis at -20°C for an average of 129.2 ± 37.3 days (mean \pm sd; range 100 – 205). Importantly, samples collected from different oestrous stages were treated with similar storage conditions which limits potential bias due to different treatments.”

We have also better acknowledged and contextualized the limitations of our study in the Discussion, section 4.2, lines 563-572:

“Previous research has shown that the volatile components in human urine degrade over time due to evaporation and bacterial activity when stored at 4°C, and more slowly when frozen, even at -80°C [99,100]. Accordingly, loss of VOCs may have occurred from the samples during storage at -20°C, prior to transfer to the chromatoprobes, although the storage at freezing temperatures likely limited this loss. Our study was outside of the tested range of storage conditions for urinary samples [99,100], so we cannot completely predict the impact on loss of highly volatile compounds involved in chemical signalling. Nevertheless, we do not expect systematic biases with respect to our conclusions regarding differences across oestrous stages because samples from different cycle phases were exposed to the same storage conditions.”

In addition, we specifically address the potential limitations of prolonged storage using chromatoprobes in our response to the next Reviewer’s comment below.

3. The manuscript also contains similar statements about storage of chromatoprobes which is equally problematic. Pg. 8 l. 35-37 indicates that chromatoprobes were stored for 450 days prior to analysis. The manuscript text indicates that these have stable retention during storage (pg. 8 l. 35-37, pg. 17 l. 46-51, pg. 20 l. 19-23), however no references, or data, are given to support this assertion. Instead, the authors cite “unpublished data” as support, without providing readers with any details on in this information. If the authors have data that these chromatoprobes can prevent VOC degradation (and/or retention biases) for over 450 days, then I strongly encourage them to publish this as a separate paper, as it is a significant finding. However, given the long-standing literature showing degradation of VOCs in a variety of media, and the extended time frame of storage in this study (129 days on average as urine, plus 450 days in a chromatoprobe), readers need to be presented with solid evidence to support the assertion that degradation is within acceptable limits. Vague references to unpublished data are not sufficient.

We thank the Reviewer for giving us the opportunity to better support our statement. We now point to previous research that has demonstrated the high shelf stability of Tenax (i.e. up to 27 months) used in thermal desorption tubes, which are used for similar applications (Woolfenden 1997). Our chromatoprobe VOC traps are even more durable because in addition to using Tenax and Carbotrap (both of which are more durable than other common materials, e.g. silicon, used in other types of probes), they also contain 1.5 mg of Carbosieve S-III. Carbosieve S-III has a higher affinity for most VOCs, and is 4-fold more absorbent than Tenax (Magnusson et al 2015), which translates into a longer shelf life. Our probes are designed this way because most of the work we do with them is in harsh tropical conditions where freezing is not always possible and travel time between sampling and freezing is measured in days, or in some situations even weeks, and it can be months to years before GC-MS analyses.

We now provide additional information on the durability of the materials used in the chromatoprobes in the Methods, lines 279-281:

“The absorbent materials used in the chromatoprobes, in particular Tenax and Carbosieve S-III, have a high and stable affinity for VOCs, which promotes a high shelf stability [44].”

In the Discussion we are now more conservative in our claims about effects on our study. We have removed the following reference to short storage and signal loss: “We are therefore confident that given the short storage time and continuous cooling, no significant amount of signal was lost.” (line 576).

We have added more information on the composition of our chromatoprobes, and improved the wording to enhance clarity, in the Discussion, lines 558-563:

“The absorbent mixture our probes are composed of (i.e. 1.5 mg Tenax TA, 1.5 mg Carbotrap, and 1.5 mg Carbosieve S-III) functions as a general, multi-purpose absorbent tailored to exceptional long-term storage, even under field conditions without access to frozen storage. The absorbent materials have a high and stable affinity for most VOCs, as demonstrated by Woolfenden [98] and Magnusson et al. [44] for Tenax and Carbosieve S-III, respectively. As a result, our chromatoprobes should have high shelf stability. In other applications, they did not

observably lose signal strength even after >3 years and with transport from remote tropical stations and no cooling [Nevo, unpublished data].”

We have also provided a summary statement which speaks of all the potential limitations of our study with regard to storage conditions, with future recommendations, lines 572-576: “We believe that given the stability of chromatoprobes, the continuous freezing applied to our samples, and similarity of treatment of samples across different oestrous stages, our conclusions are not strongly biased by the storage conditions, but systematic future examination would be useful for revealing the precise impacts of storage conditions on analytical outcomes.”

Our unpublished data support the claims based on the cited literature and we feel they do demonstrate the stability of our chromatoprobes. We provide these data for the review team (below). However, the data we have are probably not sufficient to publish as an individual paper. Below we show the (standardized) amounts of three main chemicals (styrene, ethanol, methyl isovalerate) in the headspace samples of an east African fruit species (*Balanites wilsoniana*) which were sampled over 15 months and stored at room temperature for extended periods, but analyzed on the GC-MS in one batch. Two of the three chemicals show higher amounts in the earlier samples, and one has higher amounts in the later-sampled fruits. This implies that there is no clear trend of overarching signal degradation over 15 months in conditions much, much harsher than the ones experienced by the samples used for the present manuscript. It should be noted that these are not fully standardized samples (they were ripe fruits of the same species, not the same standardized mixture), which introduces noise. Yet still, we believe that this demonstrates that samples collected using the chromatoprobes and stored at -20°C for 15 months are not likely to suffer from significant degradation that impacts our conclusions.

We believe that including this figure as a supplementary material in our submission would be off-topic, because it was part of a very different study; however, may the editor request us to include the figure in our submission, we would be willing to do so.

Woolfenden E (1997) Monitoring vocs in air using sorbent tubes followed by thermal desorption-capillary gc analysis: Summary of data and practical guidelines. *J Air Waste Manag Assoc* 47:20–36. <https://doi.org/10.1080/10473289.1997.10464411>

Magnusson R, Rittfeldt L, Åstot C (2015) Evaluation of sorbent materials for the sampling and analysis of phosphine, sulfuryl fluoride and methyl bromide in air. *J. Chromatogr. A* **1375**, 17–26. <https://doi.org/10.1016/j.chroma.2014.11.077>

4. Lastly, while the authors have clarified their sampling procedure in this revision, there is still no explanation given for how the subset of samples for odor analysis were selected. While there were clearly constraints on analyzing all samples collected, supplement table S1 suggests that selection of samples was not random. The selection procedure should be described for readers, with any pertinent notes on avoiding selection biases (conscious or unconscious).

We apologize for the lack of clarity around sample choice.

Our GC-MS machine needed to be serviced shortly into our study (after we had measured most of the samples collected on days 1-5 of the mouse study). When a GC-MS was again available we had a more limited time to complete the research. Therefore, we selected samples to best sample across the different oestrous stages for each mouse, prioritizing the samples collected more recently.

For improved transparency, we have now introduced mention of these limitations at the end of the first paragraph of section 2.3, lines 222-230:

“Our portable GC-MS needed to be serviced in the middle of our study, when we began measuring samples collected after day 5 of mouse sampling. As a consequence, and due to time constraints, we were not able to analyse all of the urine samples collected. When our equipment was returned, we prioritized the samples collected more recently, i.e. at the end of the mouse sampling, while trying to the best of our ability to sample across all oestrous stages for each mouse. In total, we analysed 90 samples by portable GC-MS (suppl. table S1). We further analysed a representative subset of these samples (i.e. 40 samples) by benchtop GC-MS to compare the methodologies for chemical analyses (suppl. table S1).”

In addition, in the hopes of improving clarity, we now present the portable GC-MS methods (section 2.3, lines 232-261) before the benchtop GC-MS methods (section 2.3, lines 263-295) in the Methods, as introduced lines 220-222 of the revised manuscript:

“Urinary VOCs were then analysed using two different methods: 1) desorption onto a portable GC-MS (PerkinElmer Torion® T-9) following extraction using a SPME fibre and 2) desorption onto a benchtop GC-MS (Agilent) following adsorption by chromatoprobe VOC traps.”

We hope this improves the logic and the flow, as the benchtop analyses were conducted on a subset of the samples analysed on the portable unit.

Following the changes mentioned above, we have further edited the paragraph describing the portable GC-MS methods by removing the first sentence (line 233): “We selected a representative subset of samples for VOC analysis using portable GC-MS.”.

We have additionally provided more detail about the selection of samples for chromatoprobe VOC trap sampling in the paragraph about the benchtop GC-MS methods, lines 263-267: “We selected a subset (40) of the urine samples that were analysed on the portable GC-MS, for benchtop analyses using an Agilent GC-MS system (suppl. table S1). We selected samples that contained greater than five drops of urine for adsorption using chromatoprobe VOC traps, as we could more reliably transfer samples that contained larger volumes of urine via pipet from the vials into the sealed sampling bag.”

Detailed comments (Please note that page numbers refer to labeled page numbers on the tracked changes version of the manuscript, not the proof PDF’s page numbers).

Pg. 7 l. 45 & Pg. 8 l. 35 The total storage times should be better clarified for each method. I assume this means that all urine was stored for an average of 129 days, while the benchtop samples were stored for an additional 450 days in a chromatoprobe. This would yield a total storage time of around 1.5 years (which is sizeable). Were the samples analyzed via portable GC-MS measured after the timeframe provided in pg 7 l. 45?

We agree. As we explain in detail in our response to Reviewer 2’s points 2 and 3 above, we have now provided more details about our storage times in our manuscript. We are grateful for the opportunity to improve the clarity. We analysed all urine samples via portable GC-MS after initially storing the urine for 129.2 +/- 37.3 days. We collected chromatoprobe samples on the same days as we performed the analyses on the portable system, and these were then transported to Germany and stored frozen for an additional 450 days until the benchtop GC-MS was available for use with our samples.

We have now made changes to our Methods and Discussion to better acknowledge the storage time in our study, both for the urine vials and the chromatoprobe VOC traps:

1) Urine storage

Lines 205-209 (Methods):

“Due to delays caused by the portable GC-MS needing to undergo servicing, we stored the vials prior to analysis at -20°C for an average of 129.2 ± 37.3 days (mean ± sd; range 100 – 205). Importantly, samples collected from different oestrous stages were treated with similar storage conditions which limits potential bias due to different treatments.”

Lines 563-572 (Discussion):

“Previous research has shown that the volatile components in human urine degrade over time due to evaporation and bacterial activity when stored at 4°C, and more slowly when frozen,

even at -80°C [99,100]. Accordingly, loss of VOCs may have occurred from the samples during storage at -20°C, prior to transfer to the chromatoprobes, although the storage at freezing temperatures likely limited this loss. Our study was outside of the tested range of storage conditions for urinary samples [99,100], so we cannot completely predict the impact on loss of highly volatile compounds involved in chemical signalling. Nevertheless, we do not expect systematic biases with respect to our conclusions regarding differences across oestrous stages because samples from different cycle phases were exposed to the same storage conditions.”

2) Chromatoprobe VOC traps storage

Lines 279-281 (Methods):

“The absorbent materials used in the chromatoprobes, in particular Tenax and Carbosieve S-III, have a high and stable affinity for VOCs, which promotes a high shelf stability [44].”

Lines 555-563 (Discussion):

“The absorbent mixture our probes are composed of (i.e. 1.5 mg Tenax TA, 1.5 mg Carbotrap, and 1.5 mg Carbosieve S-III) functions as a general, multi-purpose absorbent tailored to exceptional long-term storage, even under field conditions without access to frozen storage. The absorbent materials have a high and stable affinity for most VOCs, as demonstrated by Woolfenden [98] and Magnusson et al. [44] for Tenax and Carbosieve S-III, respectively. As a result, our chromatoprobes should have high shelf stability. In other applications, they did not observably lose signal strength even after >3 years and with transport from remote tropical stations and no cooling [Nevo, unpublished data].”

We have added a summary statement about the potential effects of storage conditions on our study outcomes. Lines 572-576:

“We believe that given the stability of chromatoprobes, the continuous freezing applied to our samples, and similarity of treatment of samples across different oestrous stages, our conclusions are not strongly biased by the storage conditions, but systematic future examination would be useful for revealing the precise impacts of storage conditions on analytical outcomes.”

Pg 7 l. 46-49 While this statement acknowledges that some degradation may occur, it does not seem to accurately portray this study (noted above), which recommended that urine be stored at -80°C (a frequent recommendation in VOC research which was not taken in the current study), and still found degradation at that colder temperature, over just a fraction of the time that samples were stored for in the current study.

We agree with the Reviewer and have revised our manuscript to better portray the cited literature and to take into account the limitations of our work in terms of storage duration, as detailed in our response to Reviewer 2’s point 2 above.

We realise aspects of this study were not in accordance with gold standard storage protocols and we wish they could have been. We were faced with limitations in access to equipment. We appreciate the opportunities to clarify these aspects and feel that despite the limitations, our

data bring valuable new discoveries, and our conclusions are supported by the data. We are pleased that we now more clearly disclose the limitations so that future researchers may benefit to the fullest degree, with the appropriate caveats. We are very grateful for the careful feedback that has allowed us to improve our manuscript.

Pg 7. l. 59 – pg. 8 l. 5 The argument here seems to be that standardizing sample volume doesn't matter because it is really urine concentration that dictates the number of VOC detected and peak area. However, the study does not appear to measure or standardize urine concentration either, so it is unclear how helpful or effective this comment is. This addition seems to undercut the reason for providing the information in Supplemental Figure S1.

We apologize for the confusion and we have edited this passage to improve clarity. We have removed reference to concentration because, as the reviewer correctly points out, this was not a feature of our study, nor is it necessary to our message.

Our new sentences (lines 212-215) reads: “Our sampling procedure is anticipated to be robust to differences in sample volume because when samples are allowed to equilibrate and are measured at the same standardised temperature (i.e. 21°C), vapor pressure should not vary with sample volume [39].”

In addition, we have removed “fully” in the following sentence, line 210: “We decided against fully standardising urine volume prior to analysis as it would have led to unnecessary loss of VOCs...”;

And we have rephrased the following sentence, lines 215-218: “To examine this, especially as there was a wide range of urine volume sampled (i.e. a 20-fold variation), we ran regressions of sample volume against the number of VOCs and their relative abundance, and verified the lack of a relationship between sample volume and these variables (suppl. figure S2).”

Pg. 8 l. 14-16 Details on the selection procedure for the subset of odorant samples that were analyzed should be provided.

Thank you. We agree and now provide details on the selection procedure for samples analysed using SPME extraction and portable GC-MS in lines 222-228, and for samples analysed using chromatoprobe VOC traps and benchtop GC-MS in lines 228-230 and 263-267.

Please also see our detailed response to Reviewer 2's point 4 above.

Pg 8 l. 35-36 This appears to contradict the literature on the storage of VOCs. Evidence that VOCs can survive 450 days in a chromatoprobe (on top of 129+ days frozen?) needs to be provided for readers. Citing past personal experience, with no data or peer-reviewed literature to support it, is not an acceptable level of rigor for publication.

We appreciate this comment and agree with the Reviewer. We now provide several citations that speak to the very long shelf life of the components of these probes. In addition, we provide for the Reviewers and Editor a figure from results produced using the same probes (in a different study) to demonstrate that the length of storage time does not systematically bias the analyses (please also see our detailed response to Reviewer 2's point 3 above).

Pg. 9 l. 3-6 Details on the subset selection procedure should be provided here as well. It is not clear what “representative” refers to (Oestrus cycle? Volume? Odors?), particularly since this sample size overlaps with, and is larger than, the subset of samples analyzed for odors via benchtop.

We agree and have now provided a more detailed explanation (Lines 222-230) and have also reorganised the Methods section 2.3 to improve clarity. Please see our response to Reviewer 2's point 4, above for more details.

Pg. 16, l. 16 Data on the “efficiency” of this method are not reported in the results; it is unclear what result this refers to. Also, the study switched collection methods part way through, so describing this method as “consistent” does not seem accurate.

We thank the Reviewer for raising this. We agree that we should be more precise in our language. We have accordingly replaced “an efficient and consistent method” with “a successful method” because it successfully allowed us to identify the successive stages of mouse oestrous (line 504).

Pg. 17 l. 46-51 No references are provided to support the assertion that chromatoprobes can preserve VOCs for over three years. If the authors have these data, I strongly encourage them to publish. Given the well documented degradation of VOCs even in ideal conditions and over shorter time frames, a vague reference to unpublished data is not particularly convincing. Likewise referring to this study's storage period of over a year as “short” does not seem to be an accurate descriptor.

We thank the Reviewer for this comment. We agree that the storage time was not “short”, so we have changed the last sentence of the argument accordingly (lines 572-576):

“We believe that given the stability of chromatoprobes, the continuous freezing applied to our samples, and similarity of treatment of samples across different oestrous stages, our conclusions are not strongly biased by the storage conditions, but systematic future examination would be useful for revealing the precise impacts of storage conditions on analytical outcomes.”

Pg. 20 l. 6-7 It is unclear what data this statement (“complex biological nature of urine...”) is referring to. Wasn't the GC-MS method the challenging issue, rather than the complexity of urine? Or is this referring to the discussion of MUP interactions? Given that MUPS and chemical interactions were not measured, it is unclear if this sentence should be included in the conclusions or should be rephrased.

We agree that this sentence introduces confusion and should not be included. We have now removed it (line 659).

Pg. 20 l. 21-23 As above, evidence or published studies need to be given to support the statement that this method produces stable retention of VOCs. Data on this are not provided in the manuscript, nor are any references provided that support this assertion. This conclusion also seems tangential to the primary study goals to compare benchtop to portable GC-MS devices and characterize VOC changes over the mouse oestrus cycle.

We agree that more data are needed to support this and we have removed the statement from the conclusion (line 666).

We thank the Reviewer for the rigorous and helpful review, which has helped us to best present our data and to not overstate our findings.

In addition to the revisions made in response to the Reviewers' comments, we have made minor edits throughout the manuscript in order to improve the clarity of our text. None of these edits (also colored in blue) are significantly changing the meaning of our text.